

# 1 Insights into the vulnerability of vegetation to tephra

# 2 fallouts from interpretable machine learning and big

# 3 Earth observation data

Sébastien Biass[1,2], Susanna F. Jenkins[1,3], William H. Aeberhard[4], Pierre Delmelle[5], Thomas
Wilson[6]
[1] *Earth Observatory of Singapore, Nanyang Technological University, Singapore*
[2] *Department of Earth Sciences, University of Geneva, Switzerland*
[3] *Asian School of the Environment, Nanyang Technological University, Singapore*
[4] *Swiss Data Science Center, ETH Zürich, Switzerland*
[5] *Environmental Sciences, Earth and Life Institute, UCLouvain, Belgium*
[6] *School of Earth and the Environment, University of Canterbury, New Zealand*
Corresponding author: Sébastien Biass (sebastien.biasse@unige.ch)
**Keywords** : Big EO data ; interpretable machine learning ; volcanic hazards; vulnerability
model; vegetation impact; natural hazards; disaster risk reduction; Google Earth Engine;

## 15 Abstract

Although the generally high fertility of volcanic soils is often seen as an opportunity, short-
term consequences of eruptions on natural and cultivated vegetation are likely to be negative.
The empirical knowledge obtained from post-event impact assessments provides crucial
insights into the range of parameters controlling impact and recovery of vegetation, but their
limited coverage in time and space offers a limited sample of all possible eruptive and
environmental conditions. Consequently, vegetation vulnerability remains largely
unconstrained, thus impeding quantitative risk analyses.
Here, we explore how cloud-based big Earth Observation data, remote sensing and interpretable
machine learning (ML) can provide a large-scale alternative to identify the nature of, and infer
relationships between, drivers controlling vegetation impact and recovery. We present a
methodology developed using Google Earth Engine to systematically revisit the impact of past



eruptions and constrain critical hazard and vulnerability parameters. Its application to the
impact associated with the tephra fallout from the 2011 eruption of Cordón Caulle volcano
(Chile) reveals its ability to capture different impact states as a function of hazard and
environmental parameters and highlights feedbacks and thresholds controlling impact and
recovery of both natural and cultivated vegetation. We therefore conclude that big EO data and
machine learning complement existing impact datasets open the way to a new type of dynamic
and large-scale vulnerability models.

## 1. Introduction

In 2015, more than 8% of the world's population lived within 100 km of a volcano that had a
significant eruption during the Holocene (Freire et al., 2019). Current trends indicate that this
exposure will increase with, for instance, the population in the two regions most exposed to
volcanic hazards (i.e. SE Asia and Central America) having doubled since 1975 (Freire et al.,
2019). Supporting up to 10% of the world's population, the fertility of volcanic soils partly
contributes to these increasing demographics (Rampengan et al., 2016, Loughlin et al., 2018).
However, farming systems remain subject to short-term negative impacts from volcanic hazards
(Choumert and Phinélias, 2018; Few et al., 2017; Phillips et al., 2019; Sivarajan et al., 2017).
Recent, modest-sized eruptions over the past decade have illustrated the large numbers of
people affected by volcanic activity, and the losses associated with impacts to agriculture, in
particular the crop subsector. For example, the 2020 VEI 4 (Volcanic Explosivity Index,
Newhall and Self, 1982) eruption of Taal (Philippines) affected ~260,000 people and caused an
estimated 63 million USD impact on agriculture (ReliefWeb, 2020), whereas the 2018 eruption
of Fuego (Guatemala), also a VEI 4, indirectly affected ~1.7 million people and caused ~58
million USD impact on agriculture (The World Bank, 2018). By comparison, a recent study by
Jenkins et al (2022) estimates that on the island of Java in Indonesia only, a VEI 4 eruption has





a 50% probability of directly affecting ≥5 million people and ~700 km$^2$ of crops, which
increases to ~29 million people and 12,000 km$^2$ of crops for an eruption of VEI 5.
The Food and Agriculture Organisation (FAO, 2018) notes how the absence of a systematic
and in-depth documentation of the impacts of natural hazards on agriculture prevents acquiring
a global understanding of their long-term direct and indirect as well as tangible and intangible
consequences. This is especially true for volcanic risk. Our current knowledge of the
vulnerability of agriculture to volcanic hazards comes from a combination of opportunistic
field-based post-event impact assessments (post-EIA; e.g., Blake et al., 2015; Le Pennec et al.,
2012; Magill et al., 2013; Phillips et al., 2019; Stewart et al., 2016; Wilson et al., 2011; Wilson
et al., 2013) and even rarer experimental studies (e.g., Hotes et al., 2004; Ligot et al., *in prep.*).
However, the generalisation of these empirical lessons is limited by two main aspects. Firstly,
eruptions are relatively infrequent but display a wide range of behaviours, each of which has
specific hazard, hazard characteristics, and impact mechanisms. Secondly, they occur over a
large variety of climates and affect various vegetation types and agricultural practices.
Damage/disruption states (DDS) derived from these data (e.g., Craig et al., 2021; Jenkins et al.,
2015; **Table 1**) have contributed to identifying critical components of vulnerability, but
currently remain too limited in time and space to allow for the development of accurate and
generalised risk models.
Satellite-based Earth Observation (EO) data, on the other hand, provide a data acquisition
framework that is both global in space and consistent in time. Missions such as Landsat,
MODIS or Sentinel now provide five decades of global EO data at a constantly increasing
spatial, temporal and spectral resolution. Monitoring of the spectral characteristics of vegetation
using these missions has been used to assess the recovery of vegetation after earthquakes (Chou
et al., 2009; Lu et al., 2012) and droughts (Rembold et al., 2019) or to derive global-scale



datasets to estimate food security (Meroni et al., 2019). In volcanic contexts, satellite imagery
has been used to capture the impact of eruptions on vegetation (de Rose et al., 2011; Marzen et
al., 2011; De Schutter et al., 2015; Easdale and Bruzzone, 2018; Li et al., 2018; Tortini et al.,
2017). Although innovative, these attempts mostly relied on single case studies, simplified
representations of hazards and never systematically investigated the range of factors controlling
the impact and recovery. The dominant limitation behind this latter point is a data processing
issue: despite the availability of an unprecedented variety of data through EO, this big EO large
dataset is associated with new challenges regarding data access, storage and processing. These
challenges have prevented the systematic investigation of the nature and the relationship
between the various processes controlling vulnerability and impact of vegetation to volcanic
hazard from a global remote sensing perspective.
However, the recent advent of cloud-based EO data storage and processing platforms paves the
way for the development of methodologies that can exploit the full potential of big EO data
(Giuliani et al., 2019; Gomes et al., 2020; Mahecha et al., 2020). Beyond providing a framework
for data-intensive research, big EO data platforms contribute to systematically extracting and
processing raw data into information and knowledge (Lehmann et al., 2020; Nativi et al., 2020;
Rowley, 2007). Over the past five years, *Google Earth Engine* (GEE; Gorelick et al., 2017) has
seen the highest increase in applications reported in the scientific literature. GEE provides
access and a computing power to process big EO data enabling reproducible, global scale
analyses (Tamiminia et al., 2020; Wang et al., 2020). GEE has been applied to aspects of natural
vegetation dynamics (Campos-Taberner et al., 2018; Kong et al., 2019; Zhang et al., 2019),
crop mapping and monitoring (Jin et al., 2019; Liu et al., 2020), land cover-land use
classification (Khanal et al., 2020), food security (Poortinga et al., 2018; Rembold et al., 2019)
and hazard mapping (Crowley et al., 2019; DeVries et al., 2020). In a volcanic context, the use
of GEE remains limited to a few applications (e.g., Biass et al., 2021; Murphy et al., 2017).





We argue that the advent of open-access cloud-based EO data platforms combined with
increasingly efficient empirical modelling approaches offer an unprecedented opportunity to
investigate the fragility of vegetation, including agricultural crops, to diverse events like
volcanic eruptions, where field studies spanning the large spatial and temporal impact spaces
are typically not possible. Here we lay the foundation of a methodology to extract previously
unexploited knowledge about the impact to, and recovery of, vegetation from past eruptions
recorded in archives of multi-spectral images. In line with the challenges identified by the FAO
(FAO, 2018), this methodology is designed to support a framework to i) unify indirect, global
with direct, *in situ* observations of impacts and ii) develop an innovative type of evidence-
based, EO-driven vulnerability model. Both factors will improve our empirical knowledge
around vegetation impacts and recovery following volcanic eruptions, supporting evidence-
based assessments for future eruptions.
Here we focus on the impacts to vegetation caused by the widespread tephra fallout deposits
from the 2011 eruption of Cordon Caulle volcano (Chile). The main steps include i)
reconstructing the relevant hazard impact metrics of the associated tephra fallout deposit using
dedicated numerical models, ii) mapping vegetation impact using time series of MODIS images
retrieved from GEE, iii) identifying and processing selected datasets and variables on GEE to
build up a big EO dataset of proxies capturing the dynamics of vulnerability in space and time,
iv) developing a flexible machine learning (ML) algorithm to trained to explain impact as a
function of the covariates and v) interpreting the model's result to investigate the nature,
importance and relationships between the different hazard and vulnerability proxies using
dedicated libraries.
**Table 1** : Damage/disruption states (DS1–5) as a function of the dry deposit thickness as hazard proxy identified
by Jenkins et al., (2014) based on literature review. DDS assume that crops are in the growing stage. Hazard
metrics include the median and interdecile deposit thicknesses inferred from expert judgement and empirical data.

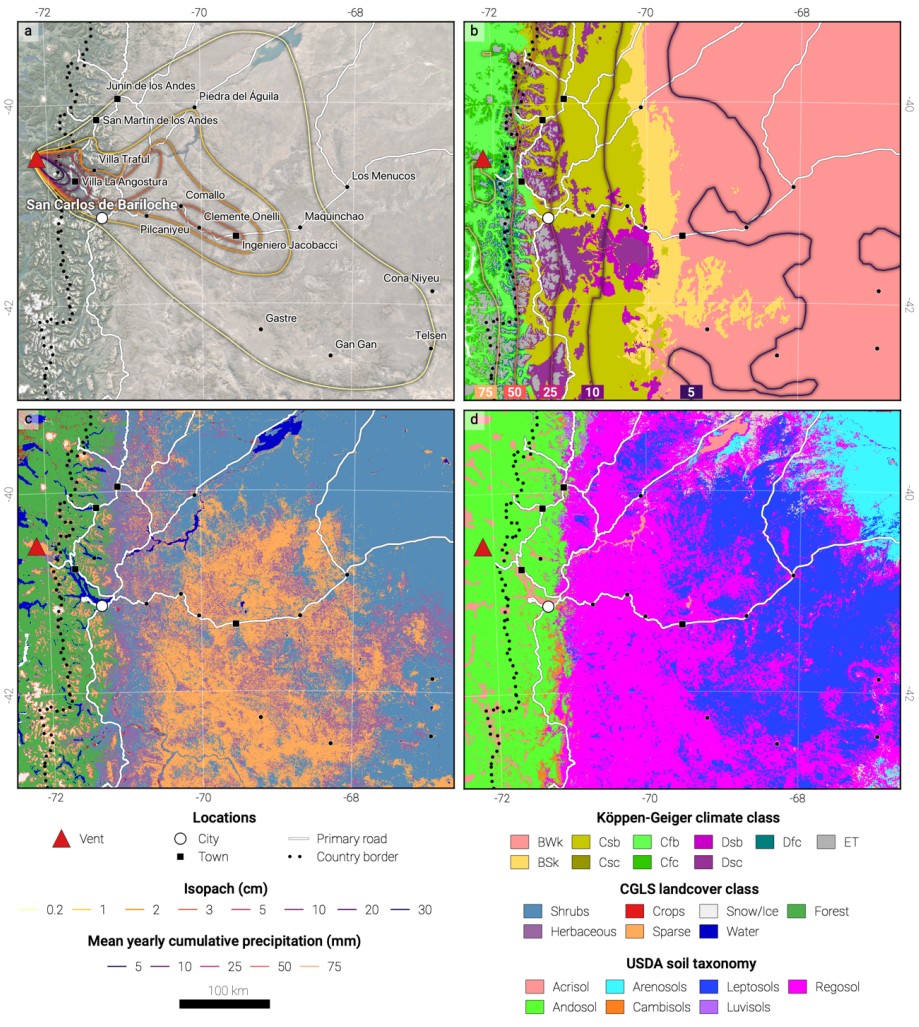


Figure 1: Overview map of the study area. a Isopach (cm) from Dominguez and Baumann (personal communication) showing lines of equal thickness of the fallout deposit for the month of June 2011. Locations are those mentioned in Elissondo et al., (2016) as being affected by tephra fall. Background is © Google Maps 2022. Roads, locations and borders are from © OpenStreetMap contributors 2021. Distributed under the Open Data Commons Open Database License (ODbL) v1.0. b Mean yearly cumulative precipitations (mm) for the period 2006-2011 inferred from ERA5. Note that these values differ from those presented in the text and in Elissondo et al., (2016) as ERA5 values represent averages over a model grid cell and time step. Background is the Köppen-Geiger climate classification of Beck et al., (2018). *BWk* - Arid, desert, cold arid, *BSk* - Arid, steppe, cold arid, *Cfb* - Warm temperate, fully humid, warm summer, *Cfc* - Warm temperate, fully humid, cool summer, *Csb* - Warm



temperate, summer dry, warm summer, *Csc* - Warm temperate, summer dry, cool summer, *Dsb* - Snow, summer
dry, warm summer, *Dsc* - Snow, summer dry, cool summer, *ET* - Polar, polar tundra. **c** Landcover classes from
the CGLS–LC1000 dataset (Buchhorn et al., 2020). **d** Dominant soil types in the study area from the SoilGrid
dataset (Hengl et al., 2017) based on the USDA soil taxonomy. All maps are projected using EPSG:32719.

## 140    2. Background

### 141    2.1.    Impact of volcanic hazards on vegetation

Explosive volcanic eruptions produce *tephra*, a generic term for pyroclasts originating from the
fragmentation of parent magma, the fraction <2 mm diameter of which is referred to as *ash.* For
sufficiently large eruptions, tephra deposits can alter the hydrology, vegetation cover and soil
properties of entire region, contributing to the perturbation of their ecosystems for months-years
(Major et al., 2016; Pierson et al., 2013). Direct negative impacts on, and the ability of
vegetation to recover from eruptions depends on complex interactions between biotic and
abiotic parameters (Ayris and Delmelle, 2012; Arnalds, 2013). Biotic parameters include the
type and composition of the vegetation, the biological legacy related to previous stresses and
the phenological state of the plant at the time of eruption. Abiotic parameters include climate
(e.g. rainfall and temperature) and environmental setting (e.g. elevation, slope, orientation)
(Crisafulli et al., 2015; Dale et al., 2005). For crops, impacts also depend on access to
technology and mitigation measures (Magill et al., 2013; Wilson et al., 2013a). Mechanisms of
adverse effects of tephra on vegetation are various, including smothering and burial, breaking
and abrasion, reduced photosynthesis, salt-induced stress and limitation of pollination (Arnalds,
2013; Ayris and Delmelle, 2012; Blake et al., 2015). Finally, impacts also depend on the
characteristics of the eruption (e.g., magnitude, intensity and style) and the deposit (i.e.,
thickness, grainsize distribution, content in water-soluble elements) (Cronin et al., 2014;
Stewart et al., 2016). Overall, tephra on crops perturbate plant phenology and may decrease or
even annihilate crop production (Ligot et al., submitted; Wilson et al., 2007).


**2.2.   Case study: The Puyehue–Cordón Caulle 2011 eruption**
On June 4 2011, a subplinian rhyolitic eruption started at Cordón Caulle volcano (CC; 40.525
S, 72.16 W; Figure 1), part of the Puyehue–Cordón Caulle volcanic complex. The eruption
began with a 24-30 h–long paroxysmal phase that gradually transitioned to low intensity tephra
emissions lasting for several months (Pistolesi et al., 2015). Reported plume heights ranged
from 9–12 km asl for the first 3–4 days, 4–9 km asl for the following week and <4 km asl after
June 14 (Bonadonna et al., 2015; Collini et al., 2013). During the first week, westerly winds
dispersed ~1 km$^3$ of tephra towards Argentina. Published isopach maps describe the deposit
thickness associated with various phases of the eruption (e.g. Bonadonna et al., 2015; Collini
et al., 2013). An unpublished report by Dominguez and Baumann (personal communication),
combining data from Bonadonna et al., (2015) and Pistolesi et al., (2015), shows the spatial
distribution of total deposit thickness for June 4–30 2011 (Figure 1a). Levels of all water-
extractable elements of the 2011 Cordón Caulle tephra were low to very-low (Stewart et al.,

174    2016).

The deposit of the CC 2011 eruption impacted three different ecosystems: from west to east,
southern Andes, Andean foothills and lowlands (Elissondo et al., 2016). These roughly
correspond to the *Warm temperate – fully humid, Warm temperate – summer dry* and *Arid*
climate classifications (Figure 1; Beck et al., 2018), respectively, each characterized by specific
assemblages of vegetation (Easdale and Bruzzone, 2018; Enriquez et al., 2021). Southern Andes
are characterized by a high elevation (mean of 2000 m asl), Valdivian temperate forest and
annual precipitations of 800–2500 mm, mainly occurring in June–August (Elissondo et al.,
2016). Andean foothills are characterized by a gradient of annual precipitation decreasing from
800 in the west to 300 mm in the east and a vegetation of grasses, shrubs, and wet meadows
covering 5–10 % of the area (Easdale and Bruzzone, 2018; Elissondo et al., 2016). The lowland
is characterized by a cold and semi-arid climate with annual precipitations of ≤300 mm. During


the six years prior to the eruption, this region experienced <160 mm of precipitation per year,
which caused regional drought conditions. Due to water availability, the rainfall gradient
strongly controls the type of farming, with pastoral farming and agriculture in Andean regions
and low intensity goat and sheep farming in the arid lowlands (Stewart et al., 2016). In addition,
regions with low precipitations experience wind erosion and remobilization of loose tephra
(Dominguez et al., 2020b; Forte et al., 2017; Wilson et al., 2011).

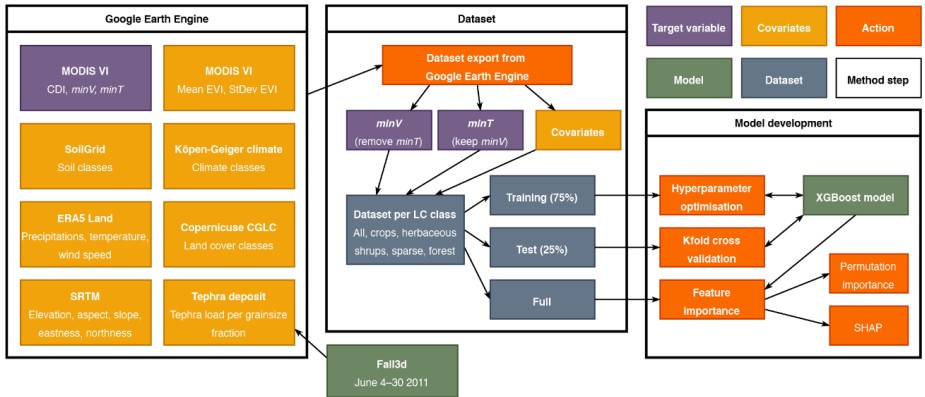


**Figure 2** : Graphical summary of the model development. Flowchart made with diagrams.net.

## 3. Material and methods

**Figure 2** summarises the conceptual steps of our new methodology to investigate the effects of
tephra on vegetation using big EO data. The aim is to train a ML model to capture vegetation
impact inferred from multi-spectral satellite images and explain it as a function of covariates
describing hazard and vulnerability. We detail the successive steps of this methodology, from
the quantification of vegetation impact (Section 3.1) and covariates (Section 3.2) to the
development, application and interpretation of the ML model (Sections 3.3–**Error! Reference s**
**ource not found.**). Throughout the paper, we refer to metrics of vegetation impact as the *target*
*variable*, whereas *feature* is used as a synonym for *co-variate* and/or *explanatory variable,* and
*instance* as a synonym for a geographic *point*.



## 3.1. Quantifying vegetation impact from remote sensing data

*In situ* assessment of vegetation (including crops) impact is typically quantified using various metrics, depending on the purpose (e.g., percentage of destroyed vegetation or yield loss; **Table 1**). We use the *Enhanced Vegetation Index* (EVI; Huete et al., 2002) as a remote sensing-based proxy for biomass production (Kong et al., 2019; Poortinga et al., 2018), and consider *impact* as a negative deviation of the post-eruption EVI signal. The EVI is retrieved from MODIS imagery (i.e., the MYD13Q1 and MOD13Q1 V6 products) generated every 16 days at a spatial resolution of 250 m. This MODIS image collection was processed on GEE.

### 3.1.1. Temporal smoothing

The MODIS EVI image collection is temporally smoothed using the median pixel value over consecutive time steps. This approach to temporal smoothing, used to reduce artefacts, was selected over filtering-based (e.g., Savitski-Golay filters) or non-parametric statistical (e.g. double logistic function) methods for two main reasons. Firstly, these methods are sensitive to the density and the signal-to-noise ratio of the time series (Cai et al., 2017). As volcanoes are vast topographic edifices, frequent clouds in their vicinity makes the application of such algorithms unstable and unreliable. Secondly, we focus on the impacts occurring at a medium-term rather than in the immediate aftermath of an eruption, where a Vegetation Index (VI) can capture signals that do not record impact (e.g., increase in soil brightness due to tephra deposit). As a result, the median value over a given time window presents the most stable and conservative smoothing method around volcanoes. We test here two-time windows of 1 and 3 months using the eruption date as a reference point.

### 3.1.2. Anomaly quantification

Multiple approaches have been developed to quantify VI *anomalies* with various purposes ranging from early warning (e.g. Asoka and Mishra, 2015; Meroni et al., 2019; Rembold et al., 2019) to index-based parametric insurance (e.g. Martín-Sotoca et al., 2019). VI anomalies have


also been used to monitor vegetation recovery after natural hazards (e.g. fires, Bright et al.,
2019; volcanic ashfall, De Schutter et al., 2015), cropping intensities (e.g. Liu et al., 2020) or
long term land degradation (Gonzalez-Roglich et al., 2019) or changes in vegetation dynamics
(Kalisa et al., 2019). We adapt the approach of Poortinga et al. (2018) as a proxy for impact of
volcanic ash on vegetation, hereafter named Cumulative Difference Index (CDI). The CDI is
computed as:

$$CDI_{ijk} = \sum_{t=1}^{t} VI_{ijk} - \overline{VI_{ij}},$$


Equation 1
where $CDI_{ijk}$ is the CDI value for pixel $i$ during the time period $j$ for year $k$, $VI_{ijk}$ is the VI
value for pixel $i$ during the time period $j$ for year $k$ and $\overline{VI_{ij}}$ is the long-term VI mean over the
baseline (averaged over 5 years prior to eruption for pixel $i$ and period $j$). $VI$ is the vegetation
index (here, EVI) and $j$ is an arbitrary time window, referring to a subset of a year. Here, $j$
considers a 1–3-month period and the baseline considers 5 years of pre-eruption conditions.
The temporal evolution of the CDI is used as the metric for impact and recovery. **Figure 3**
illustrates idealized profiles that the CDI can adopt through time. Following Equation 1, a
scenario where $CDI_{ijk}$ remains negative implies that post-eruption conditions are persistently
lower than the baseline (i.e., negative CDI slope, P1 in **Figure 3**). A CDI flattening and reaching
a zero gradient indicates a return to pre-eruption conditions (P2 in **Figure 3**). If the gradient of
the CDI slope becomes positive after the inflection point, the post-eruption biomass production
has exceeded pre-eruption conditions. If the CDI curve flattens at a negative CDI value, the
total loss in biomass due to the eruption has been partly compensated by a temporary increase
(P3 in **Figure 3**). Should the absolute CDI value become positive, the total biomass loss caused
by the eruption has been either compensated or exceeded by the gains (P4 in **Figure 3**). The




purpose of the model is to explore conditions explaining the magnitude of impact (i.e., *minV* in
**Figure 3**) and the duration to reach it (i.e., *minT* in **Figure 3**). The shape of the CDI curve after
reaching *minV* is not considered here, and *minV* for the case of P1 in **Figure 3** is the minimum
value reached after 5 years post-eruption.

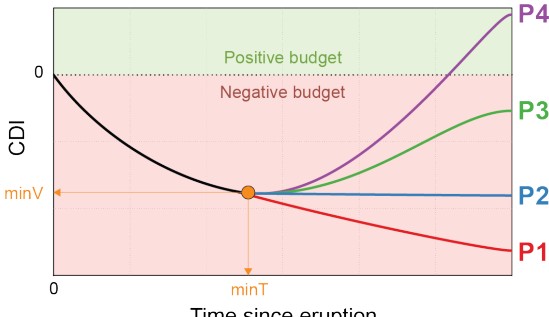


**Figure 3**: Illustration of various possible CDI profiles through time. *minV* represents the minimum CDI value
reached by a CDI profile and *minT* the duration after which *minV* has been reached. P1 represents a scenario with
a permanent degradation of the EVI. P2 represents a scenario where post-eruption conditions have returned and
remain equal to pre-eruption conditions. P3 represents a scenario where post-eruption conditions have returned
and temporarily exceeded pre-eruption conditions without compensating for the deficit caused by the eruption. P4
is similar to P3, but with post-eruption conditions sufficiently persisting to compensate and exceed the deficit
caused by the eruption.
**Table 2** : Summary of variables used in the model.
## 3.2.  Model features
Co-variates used in the model to predict the impact (**Table 2**) were chosen to capture the
relevant hazard and vulnerability parameters identified from literature (Section 2.1). Most
datasets are natively available on GEE, and others have been manually uploaded as assets. Note
that the original covariate dataset contained ~300 features. Here are presented the final set of
variables identified based on i) a minimum degree of relations in an exploratory data analysis
phase and ii) an iteration of the process of model optimisation and computation of feature
importance described in section 3.4.3 that allowed identifying and retaining the most
informative variables.
**Table 3** : Initial parameters to the Fall3d runs. For the Suzuki plume model, *A* and *λ* are the shape factor controlling
the mass distribution described by Pfeiffer et al. (2005), where *λ*=2 results in more mass distributed in the lower
portion of the plume. The *FPlume* approach (Folch et al., 2016) was solved for mass flow rate (MFR, Degruyter
and Bonadonna (2012). Two total grain-size distributions (TGSD) were tested including a field-based Gaussian
(*Md Φ* and *σ Φ* of 1.7 and 3.1, respectively; Bonadonna et al., 2015) and a model-based Bi-Weibull (modes at -
3.13 and 4.69 Φ with respective shape factors of 0.73 and 1.1 Φ and a mixing factor of 0.64; Costa et al., 2016,
Folch et al., 2021) distributions.
*3.2.1.   Deposit properties*
Deposit thickness and grain-size distribution are the two of the main physical aspects
controlling the direct impact of ashfall on vegetation (e.g., Jenkins et al., 2015). Since available
isopach maps represent only deposit thickness, we reconstructed the grainsize distribution of
the deposit associated with the June 4-30 2011 phase of the CC2011 eruption using Fall3D
v8.0.1 (Folch et al., 2021). The model was initialised using hourly atmospheric conditions
retrieved from the European Centre for Medium-Range Weather Forecasts (ECMWF) ERA5
dataset (Hersbach et al., 2020) and daily mean plume heights reported by Collini et al. (2013).
We tested several modelling schemes (**Table 3**) and compared the outputs against the isopach
in Figure 1a. For this, isopachs were interpolated using a generalised additive model and
converted to maps of tephra accumulation using a constant deposit density. We tested densities
of 1000, 2000 and 2200 kg/m$^2$ to provide a range of tephra thicknesses for each point. The
Fall3D *NetCDF* output was converted to a multiband *geotif* with each band containing mass
loads for different size fractions. Size fractions computed by Fall3D were grouped into *lapilli*
(2–64 mm), *coarse ash* (1-0.25 mm) and *fine ash* (<0.25 mm). The *geotif* was uploaded as an
asset to GEE.





*3.2.2.  Climate*
Atmospheric data were obtained from GEE using the ERA5 Land monthly averaged climate
dataset (Hersbach et al., 2020), which provides a global reanalysis of climate variables since
1981 at a spatial resolution of 0.1 x 0.1°. The total precipitation and the surface air temperature
were retrieved and their mean computed over 1, 2, 3, 6 and 12 months before the eruption. Each
variable is included both as raw values and anomalies computed as the Stand Regeneration
Index (SRI; Hope et al., 2012). As for CDI, we used a 5-years pre-eruption baseline and
normalized the closest pre-eruption value $V_{ijk}$ by the mean value over the same period in the
baseline $V_{ij}$:
$$SRI_{ijk} = \frac{V_{ijk}}{\overline{V_{ij}}}$$
Equation 2
The model also includes the wind velocity at the time of the eruption from the ERA5 Land
dataset.
In addition to atmospheric variables, the model includes Beck et al., (2018)'s updated 1-km
version of the Köppen-Geiger climate classification. The study area spans three of the five main
categories (*Arid, Warm temperate* and *Polar),* with two sub-types of the *Arid* (i.e. *Desert – hot*
*arid* and *Steppe – hot arid*) and four sub-types of the *Warm temperate* (*fully humid – warm*
*summer, fully humid – cold summer, summer dry – warm summer, summer dry – cool summer*).
A *One Hot Encoding* procedure was applied to this dataset to transform categorical labels into
the numerical values required by most ML algorithms.
*3.2.3.  Terrain*
Terrain data were obtained from the Shuttle Radar Topography Mission (SRTM; Farr et al.,
2007) using the NASA's SRTM V3 product at a resolution of ~30 m. Elevation, slope, aspect,


eastness and northness (*sine* and *cosine* of aspect, respectively) were retrieved from GEE and
used as features.

*3.2.4.  Landcover*

Landcover was obtained from Copernicus Global Land Service (CGLS) Dynamic Land Cover
map (CGLS-LC1000, Buchhorn et al., 2020), available on GEE at a spatial resolution of 100 m
yearly from 2015-2019. The landcover type is retrieved from the *discrete_classification* band
for the closest year to the eruption (here 2015). To test the impact of tephra on various types of
vegetation, we extracted the *Cultivated and managed vegetation/agriculture* class as a proxy
for cropland and the *Shrubs, Sparse* and *Herbaceous vegetation* classes (i.e., values 40, 20, 60
and 30, respectively). In addition, we extracted a composite *Forest* class comprising all classes
tagged with *Forest*. In the study area, present forest classes include *Evergreen broad leaf*, both
*Closed* (112) and *Open* (122), *Deciduous broad leaf*, both *Closed* (114) and *Open* (124) as well
as *Closed forest, mixed* (115) and *Forest, not matching any of the other definitions* (116 and
126). As for the climate classification, a *One Hot Encoding* procedure was applied to landcover
classes.

## 3.3.  Point sampling

In the study area, the vegetated landcover classes defined above account for 96% of the total
landcover, with the classes *Shrubs* (38%), *Sparse* (26%) and *Herbaceous* (17%) dominating the
total count. The *Forest* class (17%) dominates the Andean part of the study area whereas crops
represent about 1% of the region. 5000 instances were randomly sampled for each landcover
class. The target variables and covariates for all points were downloaded from GEE and stored
as a *GeoPanda* dataframe in Python.



### 3.4. Setting up the machine learning model


We developed an interpretable ML model able to process big EO data to identify the most
important variables and how they interact to cause the impact on vegetation. This amounts to a
(supervised learning) regression task; the EO data, for training and testing, include the
environmental, atmospheric, and geophysical features described above, as well as the target
variables consisting in the impact metrics. The main objective is to investigate and describe the
nature of the processes, performing out-of-sample predictions is outside of the scope of this
paper. This section introduces the ML algorithm, its optimisation and its interpretation
processes. All computations are performed using *Python 3.9* on the *Gekko* cluster of NTU's
*Asian School of the Environment*, both using CPUs and GPUs.
*3.4.1.   ML algorithm*
The main modelling challenge is to approximate complex functions mapping both *minV* and
*minT* to the various investigated features. Decision trees and related methods form a general
class of models suitable for such regression tasks. We opt for Gradient Boosted trees, a category
of decision trees that use an ensemble of so-called weak learners built sequentially to improve
prediction accuracy (Müller and Guido, 2015). Gradient Boosted trees have successfully been
applied on EO problems (e.g., Hengl et al., 2017). Here, we used the *XGBoost v.1.4.2* library,
which provides an optimised and distributed implementation of gradient boosted trees (Chen
and Guestrin, 2016).
*3.4.2.   Hyperparameter optimisation*
Gradient-Boosted trees rely on a range of hyperparameters governing the model's bias-variance
trade-off. Selected hyperparameters (Section 4.4.1) were tuned by minimising the out-of-
sample mean absolute error (MAE) computed through a 5-fold cross-validation scheme using
*Scikit-learn*'s *RepeatedKFold* and 10,000 trees. We used the *Optuna* library (Akiba et al., 2019)
optimised on a single GPU.


*3.4.3.   Model interpretation*
Gradient-Boosted trees can accommodate non-linear effects and interactions but, as for many
modern ML algorithms, come at the cost of limited interpretability. Model-agnostic
interpretation methods shedding light on black-box models are actively being developed and,
when applied on big EO data, provide a novel framework to identify and constrain the processes
driving changes through time in Earth Sciences (Batunacun et al., 2021; He et al., 2020; Sulova
and Arsanjani, 2021). Amongst these, the Shapley additive explanations (SHAP) method of
Lundberg et al., (2020), based on Shapley values (Shapley, 1956) and coalitional game theory,
decomposes any prediction from a given model as a sum of the individual effects from each
variable (Molnar, 2021). The method computes SHAP values, which quantify how a given
feature act to change a model's mean prediction. We use here SHAP values to identify drivers
of vegetation vulnerability in two ways. Firstly, the mean absolute SHAP value of a variable
across all instances indicates a relative importance amongst all features. Secondly, individual
SHAP values for a given feature and all instances provide insights into how a feature's value
influences predictions. As generalisation is not the main objective of this study, SHAP values
are computed on the full dataset. We use the *TreeExplainer* method of the SHAP library
(Lundberg et al., 2020) to explain XGBoost's prediction.
Unlike SHAP values, *permutation feature importance* ranks features based on their direct
impact on model performance (Breiman, 2001; Fisher et al., 2019). We use it as a
complementary approach to SHAP values. Permutation importance is also computed on the full
dataset using *Scikit-learn's permutation_importance* function using 10 permutations of each
variable and computing the change in the coefficient of determination $R^2$.
*3.4.4.   Modeling scheme*
A model is trained separately for each landcover class defined in Section 3.3, with one
additional model trained on all landcover classes jointly and using the landcover class as a
feature. Since *XGBoost* does not support multi-output regressions, each dataset is used as an
input for two models trained using either *minV* or *minT* as a target variable (**Figure 3**). To
include some dependence between the two impact metrics, the model predicting *minV* is trained
with *minT* removed from the features, whereas the model predicting *minT* is trained with *minV*
in the list of features.

## 4. Results

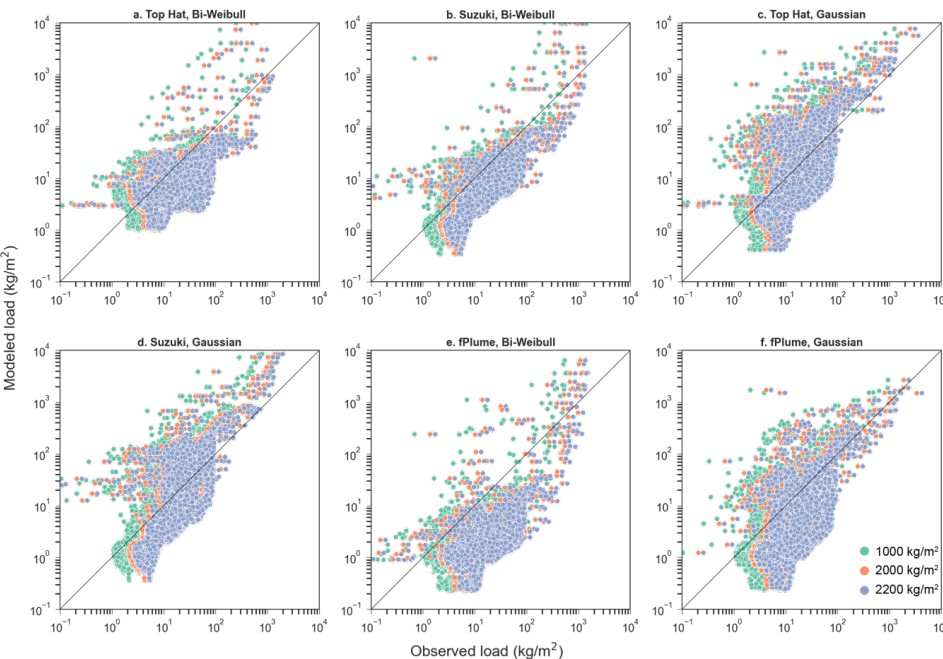


**Figure 4** : Relationship between the tephra accumulation modelled with Fall3d and inferred from isopach for the
various modelling schemes (**Table 3**). Colours consider various densities used to convert deposit thickness to mass
loads. Figure sub-labels follow **Table 3**. The black line shows a hypothetical 1:1 relationship.



## 4.1. Deposit reconstruction

To select the best Fall3d run shown in **Table 3**, 10,000 points were randomly sampled in space and used to retrieve both the modelled tephra load and the thickness obtained from interpolated isopach (**Figure 4**). Although all model runs are capturing the general trend, mismatches can be attributed to modelling issues (e.g., limitation in describing sedimentation from the plume margin or aggregation processes; Bagheri et al., 2016; Poulidis et al., 2021) and isopach interpolation using a bulk density. In the perspective of these limitations, we adopted run *b* (i.e., Suzuki plume model with a bi-Weibull grain-size distribution; **Table 3**) as it generally shows a minimum spread across the 1:1 line and provides a conservative scenario (**Figure 4**). **Figure 5** a compares the modelled load for the selected run with the isopach. The model captures both the general extend of the deposit as well as the various lobes generated as a function of variable wind conditions throughout the eruptive phase.
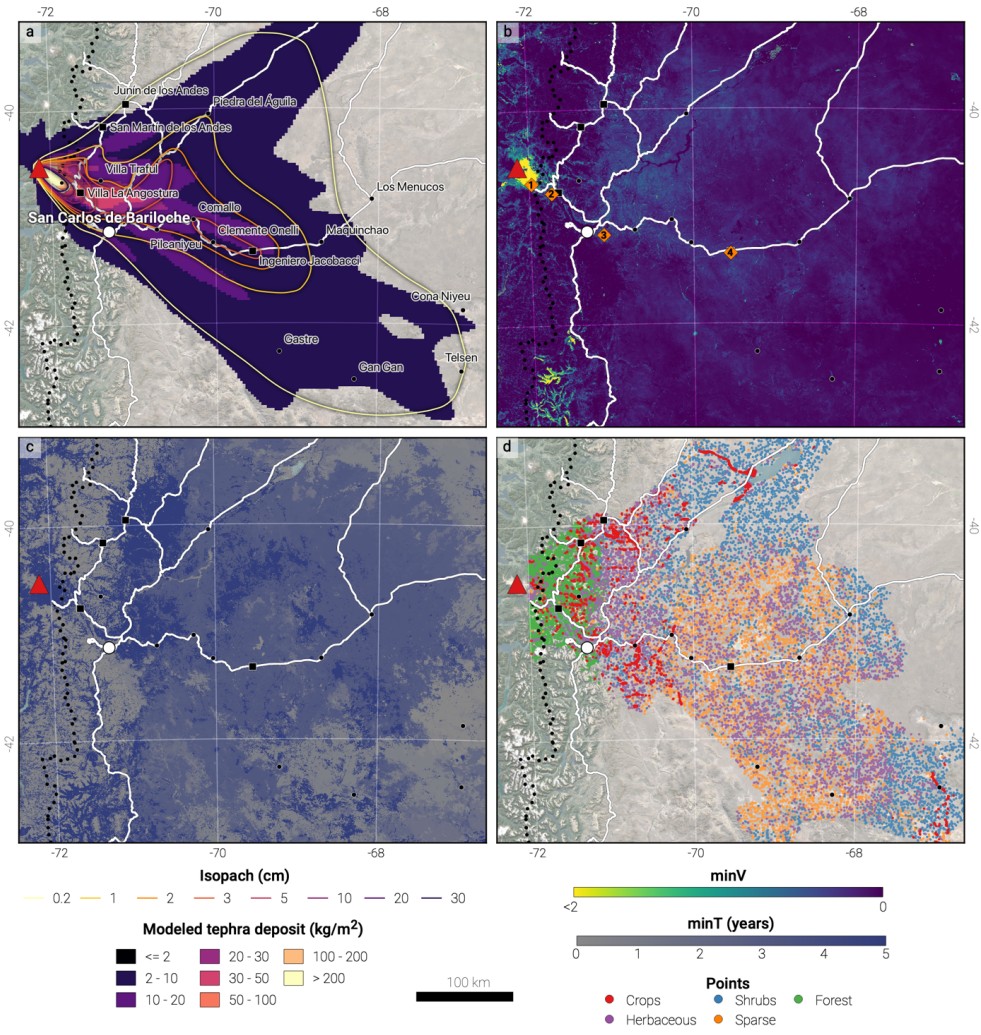

**Figure 5** : **a** Modelled load using Fall3d run b (kg/m²; **Table 3**) overlain with isopach (cm). **b** Spatial distribution of *minV*. Numbered orange diamonds are referenced in the text. **c** Spatial distribution of *minT*. **d** Dataset of points sampled in GEE coloured by their landcover class. When not specified, legend items follow Figure 1. Background is © Google Map 2022.

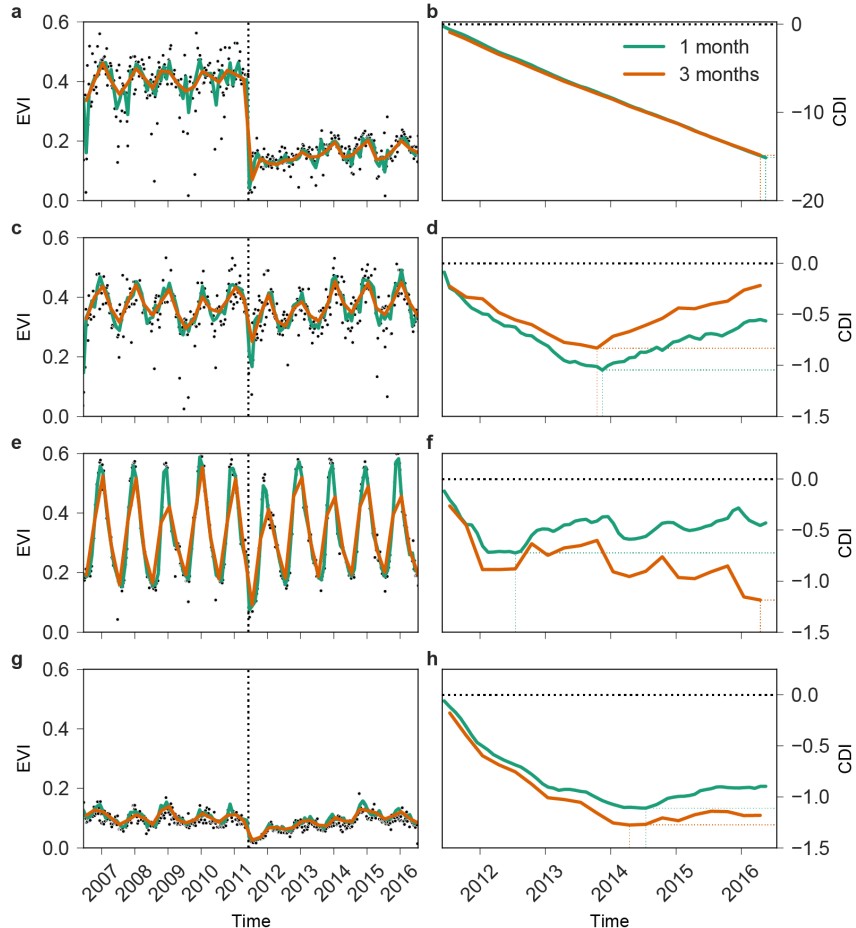

420

**Figure 6** : Time series of EVI (a, c, e, g) and monthly CDI (b, d, f, h) for the four points described in Section 4.2
and located in **Figure 5**. Black dots are raw (i.e., non-composited) MODIS data whereas green and orange lines
are composited collections using a kernel of 1 and 3 months, respectively, as described in Section 3.1. On the left
plots, the vertical black dashed line indicates eruption time. On the right plots, the horizontal black dashed line
indicates a neutral budget (**Figure 3**). Coloured dotted lines indicate the location of *minV* and *minT*.

## 4.2. Anomaly quantification

**Figure 6** shows an illustration of time series of EVI and associated monthly CDI for four

representative points in the study area (**Figure 5** b) chosen to represent the spread in tephra

accumulation and vegetation/climate types, and using compositing windows of 1 (green) and 3

(orange) months. Seasonal EVI patterns, with high values in the summer reflecting active



growth and low values in the winter reflecting plant dormancy, indicate that the eruption
occurred during a period of low growth (Elissondo et al., 2016). Point 1 (**Figure 6** a, b), located
23 km southeast of the vent, is characterized by herbaceous vegetation and a modelled tephra
load of 330 kg/m$^2$ (thicknesses of 165–330 mm when converted with deposit densities of 2000
and 1000 kg/m$^3$, respectively). The sharp drop in EVI after the eruption and the following
persistent lower values compared to the pre-eruption baseline translate into a CDI profile
showing a negative slope, which indicates that the system did not return to pre-eruptive
conditions. This observation agrees with existing DDS (**Table 1**), where accumulations ≥150
mm result in substantial vegetation destruction. Point 2, located 45 km southeast of the vent
and 7 km from Villa La Angostura consists of closed, evergreen broadleaf forest. With 40 kg/m$^2$
of tephra accumulation (thickness of 20–40 mm for the same densities as Point 1), EVI values
show a slight decrease compared to pre-eruption conditions lasting for a couple of years, after
which a general trend is observed leading to larger EVI values than the baseline (**Figure 6** c).
This translates into CDI profiles showing a negative trend for two years after the eruption, after
which a positive trend indicates better conditions compared to the baseline (**Figure 6** d). When
compared to existing DDS for forestry (**Table 1**), the modelled thickness spans damage classes
0–3, ranging from no impact to minor productivity loss. Point 3 is 112 km from the vent in the
vicinity of San Carlos de Bariloche. Classified as crops by the CGLS landcover and looking
like pastoral grazing fields from high resolution satellite imagery, it was affected by 7 kg/m$^2$ of
tephra (thickness of 3.5–7 mm; damage classes 0–3; **Table 1**). Both compositing time windows
show a reduction in EVI values for at least one season after the eruption, which translates to a
local CDI minimum about a year after the eruption (**Figure 6** e, f). Finally, Point 4 is located
240 km southeast of the vent close to Ingeniero Jacobbaci and was affected by 10 kg/m$^2$ of
tephra (i.e. 5–10 mm). Classified as herbaceous vegetation in the CGLS dataset but looking like
farmland with a mixture of pasture and crops on high-resolution satellite imagery, both EVI



and CDI profiles indicate a return to pre-eruption conditions after ~3 years, after which a
positive CDI slope indicates temporary better conditions (**Figure 6** g, h).
**Figure 6** illustrates the differences in quantifying *minV* and *minT* when using time windows of
1 and 3 months in Equation 1. A 1-month window closely follows local trends and results in
irregular CDI curves, whereas a 3-months window over-smooths local variations. Although
both approaches commonly result in similar results, Point 3 illustrates how the two windows
can induce different interpretations. We adopt a 3-months kernel for two main reasons. Firstly,
the visual comparison of the spatial distribution of *minV* and *minT* on a map shows that such
differences occur locally whilst preserving the general spatial distribution. Secondly, points
displayed in **Figure 6** are not heavily affected by cloud coverage, and the 1-month kernel does
not reflect the typical effects that clouds can induce when using such a small compositing time
window (e.g., sparse time-series, artefacts, etc.). This is generally not the case, either around
Cordon Caulle volcano where the region closer to the vent suffers too much cloud coverage to
be resolved by a 1-month kernel, or around most volcanoes around the world where large and
high edifices are often cloudy. Therefore, the 3-months kernel provides a more conservative
approach and enables reproducibility to other case studies.
**4.3.   Impact mapping**
**Figure 5** b displays the spatial distribution of *minV* in the study area. The region with the
minimum *minV* value extends up to 25 km southeast of the vent and corresponds to
accumulations of ~550 kg/m$^2$. Although conspicuous, it is impossible to unequivocally attribute
this impact to tephra fallout in proximal area where other hazards can occur (e.g., pyroclastic
density currents, lahars). Except for this region, the impact within the first 80 km east of the
vent is relatively limited, beyond which a sharp, north-south oriented decrease in *minV* values
occur. This rapid change corresponds to a change in rainfall amount, a transition from well-


developed andosols to very weakly-developed regosols and a region dominated by forests to
one dominated by shrubs and herbaceous vegetation (Figure 1; Section 2.2). In this region,
minimum *minV* values are ~-0.5 and the spatial distribution of *minV* reflects the spatial
distribution of tephra fallout. Negative *minV* values extend eastwards beyond the town of Los
Menucos, suggesting that impact occurred with accumulations ≤2 kg/m$^2$. Due to the use of a 3-
months kernel, *minT* is a discrete rather than a continuous dataset (i.e., a *minT* value of 4.5
months suggests that *minV* was reached between 3–6 months after eruption onset). The spatial
distribution of *minT* (**Figure 5** c) generally reflects *minV* and the pattern of tephra accumulation.
Note that artefacts related to non-vegetated areas are ignored (e.g., bare rock and snow-covered
mountains in the S).
**Figure 5** d shows the distribution of sampled points by landcover and selected relationships are
plotted in **Figure 7**. Although **Figure 7** a displays a general negative relationship between *minV*
and the tephra load, a simple linear relationship fails to accurately capture the variability of
impact. For *minT,* **Figure 7** b and c show how *minT* is distributed around three main modes of
tephra load and *minV*. Landcover classes that are most impacted by long *minT* values are Forests
and Herbaceous, which are the two classes the most exposed to heavy loads (**Figure 5**). Plotting
*minT* shows a distribution centred around three modes of about 400, 1000 and 1700 days
(**Figure 7** b). No clear relationship appears between *minT* and either *minV* or tephra load.

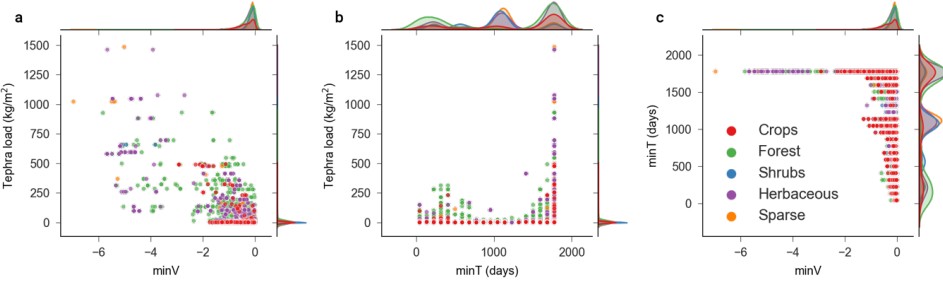





**Figure 7** : Relationship between **a** *minV* and the total tephra load, **b** *minT* and the total tephra load and **c** *minV* and
*minT* as a function of the landcover class. The marginal axes contain a kernel density estimate of the underlying
population for each landcover class. For readability all forest sub-groups are grouped.

## 4.4. ML model

**Table 4** : Summary of the trained models. The *Optimisation* columns group reports the hyperparameter values
obtained with the optimisation process. The *Model metrics* columns group reports the mean absolute error (MAE)
and the $r^2$ coefficients on both training and test datasets. The mean and the standard deviation (Std) were obtained
*by* 5-fold cross validation with three repeats.

### 4.4.1. Model performance

**Table 4** presents the results of the optimization of hyperparameters on the dataset shown in
**Figure 5** d and the associated model metrics. Hyperparameters for all model runs generally
showed a balance between parameters indicating a more (i.e., high values of alpha, lambda and
min child weight) or less (i.e., high values of max depth and learning rate) conservative model.
The MAE and $R^2$ were computed on both training and testing datasets using a cross-validation
with five folds and three repeats. We compare training and testing prediction error as an
indication of the degree of overfitting of the model. As expected, model metrics obtained on
test datasets were lower than those using training data. Based on the $R^2$ of the testing data and
*minV*, models trained on all landcover classes and on herbaceous vegetation performed well
($R^2$>0.9), followed by forests ($R^2$>0.8) and crops ($R^2$>0.7). The particularly low $R^2$ value for
sparse vegetation can be attributed to the presence of <10% vegetated cover in this class, which
is dominated by bare soil or rock. The $R^2$ values of *minT* are consistently lower than those for
*minV* and never exceed 0.6, which we partly attribute to its discrete nature.
Overall, the comparison of error metrics between testing and training sets reveal that models
trained on the various datasets have various degrees of generalisation ability, with the caveat
that the validity of the insights provided by the different models should be considered in the
perspective of their respective performances. The broadest dataset considering all landcover



classes and *minV* results in high training (0.94) and testing (0.91) $R^2$ values. We use this good
performance and similarity between both values as an indication that the model is likely not
overfitting and yields good generalisation.
**Table 5** : Ranking of feature importance computed using mean absolute SHAP values and permutation importance
for all landcover class and impact metrics. A darker cell colour indicates a stronger importance. For each column,
the 3 most important features are in bold and the 10 most important features are in red.
*4.4.2.   Feature importance*
**Table 5** summarizes feature importance for each landcover class using the mean absolute SHAP
value and permutation importance. Although some differences exist, both methods yield similar
results, thus implying that features that contribute the most to predictions (SHAP importance)
also improve the model's generalization error (permutation importance). Unless specified, this
section focuses on SHAP importance.
*EVI* and *elevation* are the two features that consistently rank in the top 10 of the most important
variables across impact and landcover. For *minT*, *minV* is the most important variable, which
suggests that both impact metrics are dependent. EVI ranks especially high, which indicates
that the mean EVI value computed over the year before the eruption provides an important
background level to the model. The variable *Lapilli* is the most important for *minV* for all
landcover classes but crops (SHAP value) and sparse (permutation importance) and ranks high
when predicting *minT* for all and the forest landcover classes.
For forests, *minV* is best predicted, in decreasing order, by lapilli, EVI and elevation, which are
respectively a deposit, a proxy for a biotic and an abiotic parameter. Note that using permutation
importance instead of SHAP importance suggests that the 3[rd] most important variable is surface
temperature, which is correlated to elevation. In parallel, *minT* is driven by *minV*, lapilli,
elevation and EVI, which indicates that the duration of impact is dominantly proportional to the
magnitude of impact and the tephra load, with additional biotic and abiotic controls. This

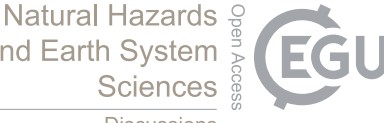
suggests that forests are potentially more resilient to moderate accumulations of ash and might
rather be prone to direct, physical impact from heavy accumulations. In comparison, the *minV*
of herbaceous vegetation is controlled by lapilli, EVI and the 6-months precipitation, which
indicates the same hierarchy of importance of deposit, biotic and abiotic parameters as for
forests, whereas *minT* is controlled by *minV,* EVI, the 3-months precipitation and fine ash.
Interestingly, this suggests that impact duration does not primarily depend on any deposit
variable, the most important of which (i.e., fine ash) is different to the parameter controlling the
magnitude of impact (i.e., lapilli). As a final example, no deposit property ranks in the top 3
variables controlling the *minV* values of crops, which include climate, EVI and the 3-months
precipitation anomaly. The first deposit parameter, fine ash, ranks 4th, which indicates that the
vulnerability of crops to ash fallout is dominantly constrained by biotic and abiotic parameters.
Fine ash ranks 5th for *minT,* which is mainly driven by *minV*, EVI and the slope, and illustrate
how abiotic parameters can potentially dominantly control impact magnitude and duration.
*4.4.3.   SHAP dependence plots*
SHAP dependence plots (Fig. 8) display, for each instance in the dataset (i.e., a point in **Figure**
**5** d), the SHAP value of a given variable as a function of its actual value. For a given instance
and a given variable, a negative SHAP values implies that the variable contributed to reducing
the predicted value compared to the mean prediction of the model. Therefore, a negative SHAP
value for *minV* implies a contribution to *increase* the magnitude of impact, whereas a negative
SHAP value for *minT* implies a contribution to *decrease* the duration of impact.
**Impact of deposit on *minV* predictions**
**Figure 8** a is the dependence plots for lapilli. With loads ≤ 60 kg/m² of lapilli, SHAP values
are contained within 0±0.1, but drastically drop for larger loads. Lapilli being dominantly
impacting the vicinity of the volcanic source, <4% of all instances are affected by
accumulations >60 kg/m² with those areas dominantly consisting of forests with additional



vegetation classified as shrubs and herbaceous (Figure 1 c). Despite limited points, **Figure 8** a
suggests stepwise decreases in SHAP values for lapilli loads of ~60, 230 and 550 kg/m$^2$. Using
a deposit density of 1000 kg/m$^3$, thicknesses of 60, 230 and 550 mm span the D1–D4 damage
states for forestry (Jenkins et al., 2014; Table 1). Using the pastoral class of Table 1 as an
analogue for shrubs and herbaceous vegetation, these accumulations suggest that, for crops,
substantial to major land rehabilitation is required before recovery. These observations confirm
the relationships between $minV$, $minT$ and the deposit load shown in Figure 7: points affected
by high lapilli loads result in $minT$ values larger than ~1300 days and an impact that persisted
for years after the eruption. These high impact metrics explain why lapilli is the most important
variable to predict $minV$. Lapilli is likely to cause a direct, physical impact from the high kinetic
energies (e.g., Blake et al., 2015; Osman et al., 2019), breakage from a static load and burial
(Arnalds, 2013; Ayris and Delmelle, 2012), which is captured as a strong anomaly by our
method and results as the most important variable. Plotting the dependence plot of lapilli for
the model trained on the generic forest landcover class (**Figure 8** b) indicates that the 2-months
precipitation anomaly contributes to further explaining the influence on the SHAP value, with
points with an anomaly <0.85 displaying lower SHAP values.





**Figure 8**: SHAP dependence plots illustrating the effect of deposit on the *minV* value predicted by the models for **a** lapilli using all landcover classes, **b** lapilli on the forest subclass and **c-j** coarse and fine ash for selected landcover classes. The hue of the points is related to additional explanatory variables. For **a, e** and **f**, the colour scheme follows Figure 1. Negative SHAP values contribute to decreasing *minV* and therefore increase impact.


Dependence plots for coarse and fine ash (**Figure 8** c, d) display similar – although less
conspicuous – drops in SHAP values for accumulations of 12 and 1.7 kg/m$^2$, respectively, with
SHAP values on average one order of magnitude smaller than for lapilli. Considering that fine
deposits are denser than coarser ones, a density range of 1000–2000 results in thicknesses of 6–
12 and 0.9–1.7 mm for coarse and fine ash, respectively, which cover the D1–D3 damage
classes for Horticultural/Arable and Pastoral agriculture (**Table 1**). Note that these thicknesses
should be regarded as minimum values as we convert here individual size fractions to total
deposit thickness. **Figure 8** e–j also shows the effect of ash for models trained on specific
landcover classes. For crops (**Figure 8** e-f), coarse and fine ash are the 10$^{th}$ and the 4$^{th}$ most
important variables, respectively. Coarse ash induces significant drops in SHAP values for
loads of 2, 4 and 10 kg/m$^2$. There is clearly an effect of fine ash on SHAP values but the
oscillatory pattern is difficult to explain for loads ≤0.5 kg/m$^2$, especially for the Csb climate
class where most crops are found (i.e., Warm temperate, summer dry, warm summer), and
probably depends on additional variables not accounted for in the model (e.g., geographic
distribution of plant-specific effects such as ash retention as a function of leaf morphology).
Beyond 1 kg/m$^2$, SHAP values are consistently negative. Coarse and fine ash are the 4$^{th}$ and the
14$^{th}$ most important variables for *minV* for herbaceous vegetation. The coarse ash shows more
negative SHAP values when associated with fine ash. Fine ash is generally beneficial for
herbaceous vegetation with low EVI values (**Figure 8** h). The most negative SHAP values are
found for high-EVI herbaceous vegetation for accumulations ≤1 kg/m$^2$ and show both positive
and negative behaviours. Since no co-variate satisfactorily explains this contrasting behaviour,
this is either due to a model artefact or to variables that are not accounted for in the model. For
shrubs (**Figure 8**i-j), coarse and fine ash are respectively the 7$^{th}$ and 12$^{th}$ most important
variables. Coarse ash shows a sharp decrease in SHAP values for loads of ~6 kg/m$^2$, beyond
which the magnitude of the negative effect increases with the lapilli load. Fine ash doesn't show
any trend or sharp break.

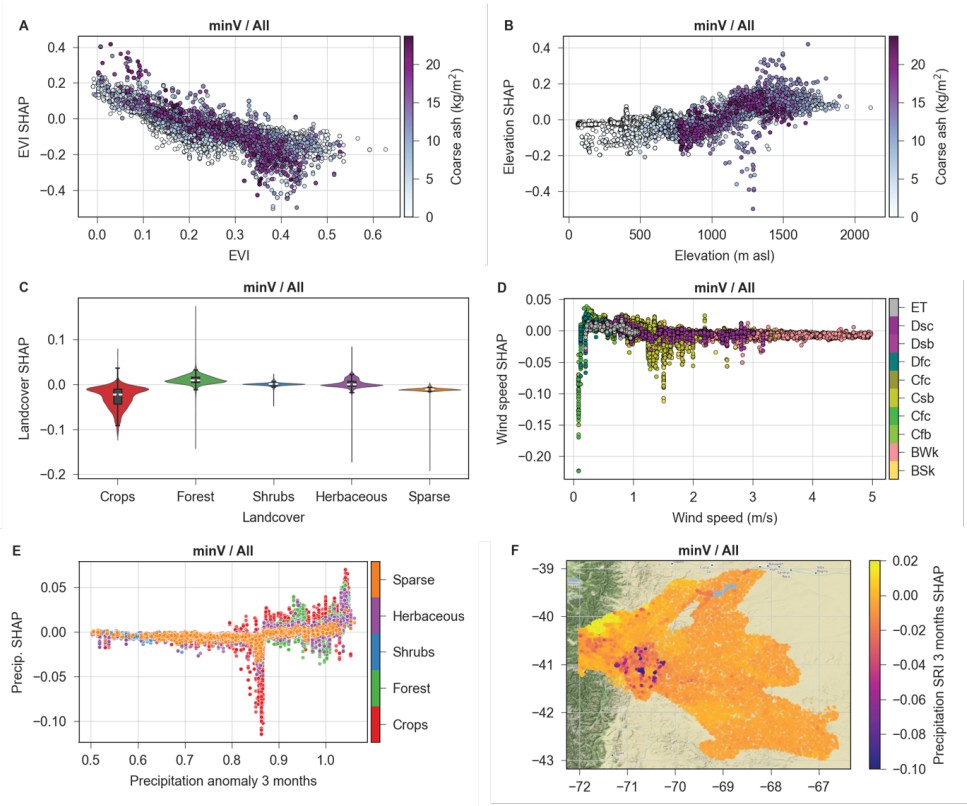


**Figure 9**: a–e SHAP dependence plots illustrating the effect of various variables on the prediction of *minV*. **a–b**
Effect of EVI (**a**) and elevation (**b**) on the SHAP value as a function of the coarse ash load. **c** Violin plot showing
the distribution of SHAP values for each landcover class with a box-and-whisker plot overlain. **d** Effect of wind
speed on the SHAP values as a function of climate. **e** Effect of the 3-months precipitation anomaly on the SHAP
value as a function of landcover. **f** Spatial distribution of 3-months precipitation anomaly SHAP values. Map tiles
by Stamen Design CC BY 3.0, map data © OpenStreetMap contributors.

**Impact of other features on the prediction of *minV***
**Figure 9** shows SHAP dependence plots for variables other than the deposit. **Figure 9** a
confirms the importance of EVI on *minV*, where all points with EVI<0.1 result in positive
SHAP values and all points with EVI>0.3 result in negative SHAP values. This observation is


partly a consequence of the use of Equation 1, where the value of $VI_{ijk} - \overline{VI_{ij}}$ is generally larger
for higher EVI values. **Figure 9** a also suggest a dependence of this relationship on the load of
coarse ash, which slightly increases SHAP values for low EVI, but decreases them for higher
values. Elevation is the 3[rd] important feature for predicting $minV$ and shows a breakpoint at an
altitude of ~1000 m asl (**Figure 9** b), below which SHAP values are dominantly negative.
Above this elevation, SHAP values are generally positive, regardless of the intensity of ash
accumulation. Landcover, the 7[th] most important feature, indicates that crops dominantly
contribute to increasing impact in the model (**Figure 9** c). Sparse vegetation also has a negative
but less pronounced effect on SHAP values, whereas shrubs and herbaceous vegetations have
a neutral effect. The SHAP values of forests tend to reduce the impact, which corroborates the
higher resilience of trees to tephra fallout (**Table 1**).
Wind and precipitation partly control the residence time of ash on leaves and therefore the
impact (Ayris and Delmelle, 2012). Although variables used here only consider pre-eruption
atmospheric conditions, they are indirectly used as indicators for post-eruption patterns. The
impact of wind speeds on SHAP values shows breakpoints at 0.2 and 1.2 m/s. SHAP values are
strongly negative below 0.2 m/s, generally positive up to 1.2 m/s and generally negative above
(**Figure 9** d). This supports the idea that wind contributes to reducing the residence time of ash
on leaves, but the aeolian remobilization of ash at higher wind speeds can negatively impact
vegetation (e.g., Arnalds, 2013; Craig et al., 2016b; Elissondo et al., 2016; Wilson et al., 2011).
Although depending on additional parameters (e.g., surface roughness, ash properties, soil
humidity, rainfall intensity), an empirical value for onset of remobilization of 0.4 m/s has been
used in the literature and agrees with our results (e.g., Folch et al., 2014; Liu et al., 2014).
Leadbetter et al., (2012) observed that ash resuspension is suppressed if precipitation rates
exceed 0.01 mm/h, and our model indicates that most negative SHAP values occur for relatively
dry climates. The most important precipitation variable for predicting $minV$ with all landcover

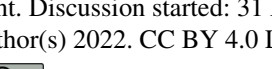



classes is the precipitation anomaly computed over 3 months before the eruption, which mostly
shows a negative anomaly (i.e., anomaly<1; Table 5; **Figure 9** e). This precipitation anomaly
shows a clear break at a value of 0.87, for which SHAP values are dominantly negative below
and positive above. Above a value of 1, SHAP values increase. **Figure 9** e shows a negative
peak in SHAP values between an anomaly of 0.85–0.87 across all landcover classes but stronger
for crops. Plotting SHAP values on a map (**Figure 9** f), the spatial clustering of negative SHAP
values corresponds to the location of crops between San Carlos de Bariloche and Comallo
(**Figure 1**). No variable unequivocally explains this spatial clustering.

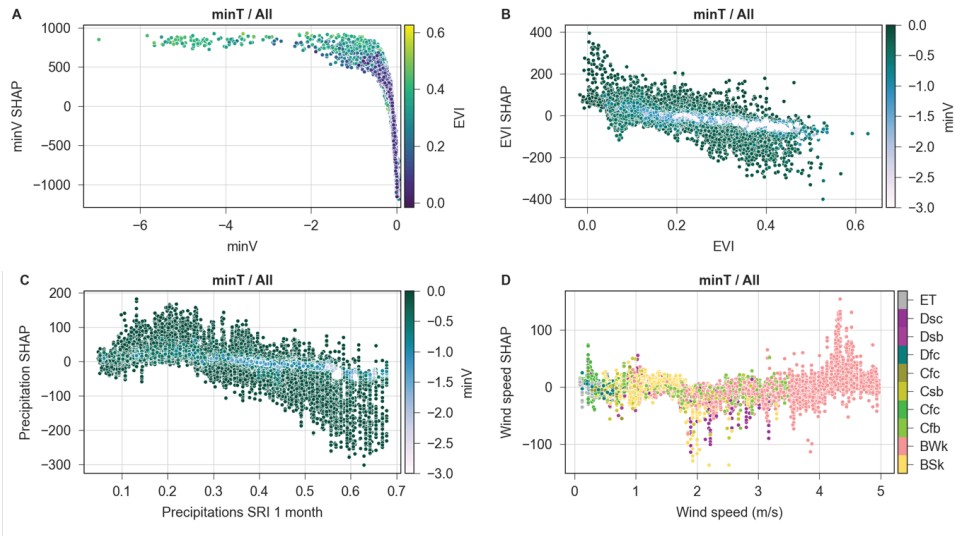


Figure 10: SHAP dependence plots for *minT* showing the effect on the SHAP value from **a** *minV* as a function of
EVI; **b** EVI as a function of *minV*; **c** 1-month precipitation anomaly as a function of *minV* and **d** wind speed as a
function of climate. Negative SHAP values contribute to decreasing *minT* and therefore decrease impact the
duration for reaching *minV*.
**Features driving *minT***
With a mean absolute SHAP value >7 times larger than any other variable, *minV* is by far the
most important for predicting *minT* (Figure 10 a), with a cut-off between positive (i.e.,
increasing the value of *minT*) and negative (i.e., decreasing *minT*) at a *minV* value of ~0.15. The



effect of EVI on *minT* is the opposite of *minV* (**Figure 9** a): although high EVI values tend to
increase the impact magnitude (lower *minV*), they generally contribute to reducing the impact
duration (i.e., Figure 10 b). Interestingly, this trend disappears as *minV* increases. This can be
explained by the fact that points affected by high *minV* values in Figure 10 b are associated with
relatively high *minT* values (**Figure 7**; Figure 10 a). These points are associated with damage
classes suggesting land retirement, and their recovery is therefore independent of the pre-
eruption EVI level. The 1-month precipitation anomaly is the 5[th] most important variable for
*minT* (Figure 10 c), and SHAP values are mostly positive below an anomaly of 0.3 and mostly
negative above 0.5. As for EVI, high *minV* values are less sensitive to the general trend. Finally,
Figure 10 d shows the effect of the wind speed at the time of eruption on *minT* as a function of
the climate. Wind speeds >4 m/s considerably increase *minT*, especially in an arid climate (i.e.,
BWk) where the vegetation is mostly shrubs, herbaceous and sparse. Points with positive SHAP
values at wind speeds >4 m/s are characterized by accumulations of fine ash >0.5 kg/m$^2$. In
contrast, points with minimum SHAP values between wind speeds of 1.8–2.8 m/s correspond
to crops close to Piedra del Aguila and show fine ash loads <0.5 kg/m$^2$.
## 5. Discussion and perspectives
The proposed methodology provides a new framework to systematically assess the vulnerability
of vegetation to tephra fallout as a dynamic, multi-variate problem. Its application to the CC
2011 eruption highlights how big EO datasets and interpretable machine learning could help
acquiring a new knowledge from tens to hundreds of understudied eruptions recorded in
archives of multispectral images. This approach aligns with FAO's objective of gaining a global
understanding of vegetation vulnerability through the systematic study of their impacts and, in
turn, contributes to various Sustainable Development Goals (SDGs 2.4, 13.1, 15.3). Specific to
volcanic risk, this is the first effort to provide a large scale, quantitative basis to estimate the



impacts of explosive volcanic eruptions on food production. On a longer time-scale and large
spatial scale, this is the first step towards tackling the unaddressed *black elephant* event that is
the risk of future large eruptions on food security (Lin et al., 2021).
**Validation and causal inference of impact mechanisms**
Our methodology focuses on impact mechanisms either occurring from the direct action or
arising from interactions between physical properties. Since we neglect the impact from water
leachable elements (e.g., Stewart et al., 2020), the approach is more suited to dominantly
magmatic events rather than eruptions with a significant hydrothermal component. Impact
patterns captured by our methodology are corroborated by lessons learned from empirical post-
EIA and experiments. For CC 2011, the model suggests that, except for points subjected to
destruction from large tephra loads, various biotic and abiotic variables tend to have a more
critical control on both impact magnitude and impact duration than deposit properties (**Table**
**5**). SHAP dependence plots for deposit properties (e.g., **Figure 8** a–e) identify similar tephra
thresholds as those identified in existing DDS (**Table 1**). Nevertheless, recent findings from
size distribution, ranging from physical impact for large lapilli to a reduction of light
interception from fine ash leading to a decrease in photosynthesis (e.g., Ligot et al., submitted;
Ligot et al., in prep). DDS must therefore consider other hazard impact metrics than only tephra
thickness, and Fig. 8–10 are the first attempt towards this objective. The method is also able to
capture impacts arising from interaction between other parameters than deposit properties. For
instance, **Figure 9** d suggests that the model captures the general relationship between presence
of ash, precipitation (inferred from climate) and wind speed in controlling the impact from
aeolian remobilisation. This demonstrates the ability of the model to identify complex and
dynamic processes, and cross-validating thresholds inferred from the model with values from
existing post-EIA and experiments provides a systematic framework to generalize observations



made at different scales (Dominguez et al., 2020a; Forte et al., 2017; Leadbetter et al., 2012;
Liu et al., 2014).
Despite these observations, methodologies for interpretable ML should be carefully used when
attempting to infer causality from correlations/associations. Suggestions of causality are
currently restricted to effects that rely on phenomena that have been either witnessed in the field
or experiments. Other variables considered in our dataset show conspicuous and complex
patterns that we are unable to explain (e.g., **Figure 8** f, **Figure 9** e). Such patterns have two
possible explanations (or a combination of both): either the model fails to accurately capture
the underlying relationship between feature and target variable, or the relationship is
complicated by other factors (e.g., feature interactions, confounding variables), including
unobserved ones. Investigating which association captures true causality therefore requires the
development of synergies between various relevant disciplines (e.g., physical volcanology,
ecology, soil sciences, disaster risk reduction). The development and adaptation of existing
causal inference methods in Earth Sciences to investigate a system's causal interdependencies
is an active topic of research (Runge et al., 2019).
**Towards a model for agricultural crops and food production**
The methodology currently relies on the CGLS-LC100 land cover dataset do distinguish
between natural vegetation and agriculture. We focus here on agricultural crops which, despite
representing ~1% of the study area, show the highest vulnerability to tephra fall (**Figure 9**).
Note that although pastoral crops are included in the *Herbaceous vegetation* class in CGLS-
LC100, it is impossible to distinguish between natural and managed grassland (Buchhorn et al.,
2020). Post-EIA on agricultural impacts have demonstrated how agriculture vulnerability
depends on various factors that are not included in our model, including some of socio-
economic nature (Blake et al., 2015; Magill et al., 2013; Phillips et al., 2019; Wilson et al.,
2013, 2007; Ligot et al., submitted) that reflect specific challenges associated with different


farming activities (e.g.,  pastoral versus horticultural, intensive versus subsistence farming).
Although future evolutions of the CGLS-LC100 dataset will possibly include finer sub-
definitions of the crops class (e.g., irrigated versus rainfed cropland, farm size; Buchhorn et al.,
2020), the methodology currently considers all agricultural crops as a uniform system.
Despite this limitation, the proposed methodology nevertheless follows impact mapping
techniques implemented in several other approaches for vegetation and food security mapping
and monitoring (e.g., Meroni et al., 2019; Poortinga et al., 2018; Rembold et al., 2019), but
differ in their fundamental purposes. To our knowledge, we provide here the first attempt to
combine numerical modelling, big EO data and ML into a framework to re-analyse and extract
new knowledge from data recorded in decades of remote sensing images as the basis for a new
type of evidence-based vulnerability model. However, several steps are required for future
evolutions of our approach to inform quantitative risk assessments on food production and
security. Amongst them, future iterations of the methodology will focus on achieving:

1.  More applications of the model to various types of climates, eruptions and sampling

different relationship between eruption date and phenological cycle in order to improve

its generalisation;

2.  Comparison, validation and scaling of the EVI-based impact metrics with other impact

estimates, either based on field interviews (e.g., yield loss), mapping (e.g., percentage

of destroyed or damage vegetation) or other indirect proxies for physical processes (e.g.,

Gross and Net Primary Productivity).

**Caveats and future research**
Below are future challenges and possible improvements of the method.

1.  The methodology takes advantage of datasets available on GEE (**Table 2**) and combines

datasets of different nature, spatial and temporal resolutions. This discrepancy affects


the accuracy of the model, and future development will explore a balance between the
spatial and temporal resolutions of all datasets. Specifically ERA5 data will be
reanalysed using mesoscale atmospheric models (e.g., Skamarock et al., 2019) at a
resolution consistent with other datasets;
2. An inherent and inevitable dependency exists between the various datasets; some are of
ecological nature (e.g., climate is a function of precipitation and temperature) whereas
other are geographic coincidences (e.g., lapilli dominantly affect the Cfb climate class,
Figure 1). Further work is necessary to explore how these dependences influence model
prediction and interpretability;
3. The methodology currently relies only on pre-eruption values for covariates. In order to
capture the evolution of post-eruptive aspects (e.g., ash residence on vegetation surface
as a function of wind and precipitations), future applications of the model will include
post-eruption variables in the training process;
4. Despite providing a satisfactory accuracy, other algorithms and models than gradient
boosted regression trees allowing multi-output predictions must be explored to model
*minV* and *minT* jointly;
5. ML models used in EO applications rarely accommodate spatial (and spatio-temporal)
dependence. Accounting for these is necessary for reliable (causal) inference and
uncertainty quantification. We plan to investigate the use of Gaussian processes, among
others, to capture any residual spatial dependence.

## 6. Conclusion

We developed a methodology to remotely quantify impact through a combination of big EO
data, interpretable ML and physical volcanology as a first step towards the development of a
framework to identify, quantify and generalize key variables driving the impact of vegetation

https://doi.org/10.5194/nhess-2022-79




after an eruption. The methodology is designed to provide a high-level and complementary
perspective to dedicated studies of the various disciplines involved in the characterization of
the vulnerability and impact of vegetation and crops to natural hazards, and has the potential to
enhance the development of new synergies between the different actors and stakeholders
involved in this specific facet of risk.
Based on the application of the methodology to the 2011 eruption of Cordon Caulle, the main
conclusions are:
- Both the magnitude and the duration components of impact captured by the processing
of MODIS satellite imagery reflect the geometry of the deposit (**Figure 5**);
- The methodology provides a systematic approach to identify the nature of the most
important variables controlling the final impact metrics. The forest landcover class is
mostly controlled by deposit properties (e.g., lapilli accumulation), whereas the crops
landcover class predominantly depends on biotic and abiotic parameters;
- Interpretable machine learning methods provide insights into the nature of impacts. For
instance, forests appear to be impacted by a direct physical impact caused by heavy
accumulations;
- Across landcover classes present in the study area, SHAP dependence plots suggest that
forest and crops are the most and the least resilient vegetation classes to tephra
accumulation, respectively (**Figure 9** c);
- The interpretation of SHAP dependence plots for deposit properties of the different
landcover classes (**Figure 8**) are in good agreement with thresholds for existing DDS
inferred from post-event impact assessments (**Table 1**), which further reinforces the
validity and usefulness of our approach.



**Author contribution**


SB designed the project, elaborated the methodology and wrote the Python library with inputs
from all co-authors on aspects of volcanic risk (SFJ, TW), interactions between tephra deposits
and vegetation (PD) and data science (WHA). All authors contributed to the manuscript.
**Competing interests**
The authors declare that they have no conflict of interest.
**Acknowledgements**
We are grateful to Edwin Tan and EOS/ASE's HPC for support on the Gekko cluster, to Lucia
Dominguez for providing isopach maps, to Jan Peuker for his patience and advice for the
development of ML modelling strategies and to Oege Dijk for developing the
*explainerdashboard* library. This work was supported by the National Research Foundation
Singapore and the Ministry of Education—Singapore under the Research Centres of Excellence
initiative (SB, SJ).

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




| Code: | DS0 | DS1 | DS2 | DS3 | DS4 | DS5 |
|---|---|---|---|---|---|---|
| **Description** | No damage | Disruption to harvest operations and livestock grazing of exposed feed | Minor productivity loss: less than 50 %/crop | Major productivity loss: more than 50 %/crop; Remediation required | Total crop loss; Substantial remediation required | Major rehabilitation required/ Retirement of land |
| **Agriculture type** | | | | | | |
| **Horticultural/Arable** | 0 mm (0-20 mm) | 1 mm (0.1-50 mm) | 5 mm (1-50 mm) | 50 mm (1-100 mm) | 100 mm (25-200 mm) | 250 mm (100-400 mm) |
| **Pastoral** | 0 mm (0-20 mm) | 1 mm (0.1-50 mm) | 25 mm (1-70 mm) | 60 mm (20-150 mm) | 100 mm (30-200 mm) | 250 mm (100-400 mm) |
| **Paddies** | 0 mm (0-50 mm) | 3 mm (0.1-50 mm) | 30 mm (1-75 mm) | 75 mm (20-300 mm) | 150 (75-300 mm) | 250 mm (100-500 mm) |
| **Forestry** | 0 mm (0-75 mm) | 5 mm (0.1-75 mm) | 200 mm (20-300 mm) | 1000 mm (100-2000 mm) | 1500 mm (100-2000 mm) | N/A |

**Table 1** : Damage/disruption states (DS1–5) as a function of the dry deposit thickness as hazard proxy identified by Jenkins et al., (2014) based on literature review. DDS assume that crops are in the growing stage. Hazard metrics include the median and interdecile deposit thicknesses inferred from expert judgement and empirical data.




| Data provider | Variable | Description | Resolution |
|---|---|---|---|
| **MODIS** | minV | Target variable for magnitude of impact | |
| | minT | Target variable for timing of impact | |
| | EVI | Mean EVI value averaged over 1 year of pre-eruption data | |
| | EVI_stdDev | Standard deviation of EVI value averaged over 1 year of pre-eruption data | 250 m |
| **Fall3D** | lapilli | Lapilli mass load (kg/m2) | |
| | coarse_ash | Coarse ash mass load (kg/m2) | |
| | fine_ash | Fine ash mass load (kg/m2) | 0.033 deg. |
| **SRTM** | elevation | Terrain elevation (m asl) | |
| | slope | Terrain slope (degrees) | |
| | aspect | Terrain aspect (degrees) | |
| | northness | Cosine of aspect | |
| | eastness | Sine of aspect | 90 m |
| **ERA5** | total_precipitation_$n$ | Total precipitation (m) | |
| | total_precipitation_SRI_$n$ | Anomaly in total precipitation | |
| | temperature_2m_$n$ | Air temperature (°K) at a 2 m elevation | |
| | temperature_2m_SRI_$n$ | Anomaly in air temperature | |
| | wind_10 | Wind speed (m/s) at a 10 m elevation | 0.1 deg. |
| **Copernicus** | landcover | Copernicus global land cover layer | 100 m |
| **Other** | climate | Köppen climate classification | 1000 m |
| | soil | Soilgrid | 250 m |

**Table 2** : Summary of variables used in the model.

Note: The total_precipitation and temperature_2m variables are calculated for $n$ = 1, 2, 3, 6 and 12 months


| Run | Plume model | Plume param. | TGSD |
|-----|-------------|--------------|------|
| a | Top Hat | Thickness = 2000 m | Bi-Weibull |
| b | Suzuki | A=4, L=5 | Bi-Weibull |
| c | Top Hat | Thickness = 2000 m | Gaussian |
| d | Suzuki | A=4, L=5 | Gaussian |
| e | Fplume | Solved for MFR | Bi-Weibull |
| f | Fplume | Solved for MFR | Gaussian |

**Table 3** : Initial parameters to the Fall3d runs. For the Suzuki plume model, $A$ and $\lambda$ are the shape factor controlling the mass distribution described by Pfeiffer et al. (2005), where $\lambda$=2 results in more mass distributed in the lower portion of the plume. The *FPlume* approach (Folch et al., 2016) was solved for mass flow rate (MFR, Degruyter and Bonadonna (2012). Two total grain-size distributions (TGSD) were tested including a field-based Gaussian ($Md\ \Phi$ and $\sigma\ \Phi$ of 1.7 and 3.1, respectively; Bonadonna et al., 2015) and a model-based Bi-Weibull (modes at -3.13 and 4.69 $\Phi$ with respective shape factors of 0.73 and 1.1 $\Phi$ and a mixing factor of 0.64; Costa et al., 2016, Folch et al., 2021) distributions.




| LC | Impact | Optimisation | | | | | Model metrics | | | | | | | |
| --- | --- | --- | --- | --- | --- | --- | --- | --- | --- | --- | --- | --- | --- | --- |
| | | Max depth | Learning rate | Alpha | Lambda | Min Child Weight | Training | | | | Testing | | | |
| | | | | | | | Mean MAE | Std MAE | Mean R2 | Std R2 | Mean MAE | Std MAE | Mean R2 | Std R2 |
| Optimisation range | | 4–12 | 0.005–0.05 | 0.01–10 | 1e-8–10 | 10–1000 | | | | | | | | |
| All | minV | 11 | 0.037 | 0.065 | 5.833 | 17.548 | 0.046 | 0.001 | 0.936 | 0.013 | 0.061 | 0.003 | 0.906 | 0.029 |
| | minT | 12 | 0.040 | 0.133 | 0.396 | 12.971 | 194.874 | 5.811 | 0.700 | 0.014 | 251.854 | 8.498 | 0.577 | 0.023 |
| Crops | minV | 10 | 0.046 | 0.094 | 8.576 | 10.157 | 0.062 | 0.006 | 0.847 | 0.042 | 0.100 | 0.010 | 0.707 | 0.053 |
| | minT | 12 | 0.045 | 0.216 | 0.157 | 15.980 | 202.460 | 16.591 | 0.651 | 0.039 | 304.758 | 28.692 | 0.470 | 0.085 |
| Herbaceous | minV | 12 | 0.050 | 0.116 | 1.274 | 21.702 | 0.040 | 0.004 | 0.955 | 0.043 | 0.053 | 0.008 | 0.907 | 0.069 |
| | minT | 12 | 0.043 | 0.525 | 0.0005 | 10.032 | 171.563 | 11.755 | 0.716 | 0.034 | 216.754 | 17.310 | 0.586 | 0.061 |
| Shrubs | minV | 11 | 0.046 | 0.094 | 0.0004 | 39.336 | 0.042 | 0.005 | 0.671 | 0.162 | 0.046 | 0.008 | 0.566 | 0.206 |
| | minT | 12 | 0.050 | 0.048 | 0.008 | 40.583 | 189.501 | 14.768 | 0.593 | 0.055 | 207.409 | 19.116 | 0.515 | 0.073 |
| Sparse | minV | 10 | 0.050 | 0.073 | 1.949 | 67.089 | 0.039 | 0.003 | 0.733 | 0.239 | 0.049 | 0.009 | 0.428 | 0.259 |
| | minT | 10 | 0.047 | 0.284 | 0.001 | 22.610 | 225.702 | 11.477 | 0.459 | 0.060 | 245.865 | 19.722 | 0.386 | 0.084 |
| Forest | minV | 10 | 0.049 | 0.123 | 2.117 | 10.999 | 0.075 | 0.005 | 0.894 | 0.030 | 0.096 | 0.008 | 0.872 | 0.045 |
| | minT | 11 | 0.049 | 0.012 | 0.023 | 16.667 | 253.507 | 15.858 | 0.669 | 0.034 | 332.919 | 18.864 | 0.543 | 0.041 |

**Table 4**: Summary of the trained models. The *Optimisation* columns group reports the hyperparameter values obtained with the optimisation process. The *Model metrics* columns group reports the mean absolute error (MAE) and the $r^2$ coefficients on both training and test datasets. The mean and the standard deviation (Std) were obtained by 5-fold cross validation with three repeats.



**Table 5**: Ranking of feature importance computed using mean absolute SHAP values and permutation importance for all landcover class and impact metrics. A darker cell colour indicates a stronger importance. For each column, the 3 most important features are in bold and the 10 most important features are in red.