# Peer review of "Insights into the vulnerability of vegetation to tephra"

_Natural Hazards and Earth System Sciences, 2022_

## Referee Comment (RC1)

**Review of "Insights into the vulnerability of vegetation to tephra fallout from interpretable machine learnig and big Earth observation data"** By Biass et al.

I read with extreme interest the manuscript submitted by Biass et al. for publication in NHESS. The manuscript presents an innovative approach to detect and characterize the impact of volcanic hazards, and specifically ash fallout, on vegetation using Earth observation time series analysed through Google Earth Engine, as well as an advanced machine learning approach to identify factors controlling the magnitude and duration of this impact. This research is ground-breaking as it proposes a new approach to tackle the highly relevant so far poorly quantified issues of vegetation vulnerability. The pilot study of the Cordon Caulle 2011 eruption illustrate the potential of the proposed approach and highlight a series of potential relationships that requires further attention.

The manuscript is very dense and rather technical, as it introduces complex methods, but the authors succeed in most part to make the approach intelligible also for non-technical expert and highlight some interesting elements of interpretation. The manuscript could however be of broader interest by highlighting the novelty and relevance of the approach for the study of impact of other hazards on vegetation (floods, fire… ?).

The manuscript is very well written and richly illustrated. All the figures appear as essential to support the findings of the paper. I consider that this research is of great interest for the community working on volcanic hazards and risk, but also to other earth scientists interested with the interplay of hazard and vegetation. The manuscript would benefit from minor revisions in place to further clarify some elements of the methods or some choice taken by the authors, as summarized hereafter.

Minor comments:

- Abstract line 21 – abstract only refer to 'eruption'; as the paper focus on impact from ash, best to mention 'vulnerability to ash fallout/explosive eruptions'.
- Line 32: "complement existing impact datasets AND open"
- Line 71: 'missions such as Landsat, Modis, Sentinel now provides five decades' – only Landsat mission provides five decades of consistent datasets, much less for the other mentioned ones.
- Line 118: 'to trained to explain' – check English
- Figure 1a: label the thickness on isopach lines (as done in Fig. 1b), the colours of isopach lines are not sufficiently clear to link with caption.
- Line 172-74: the reference to 'water-extractable' elements might be cryptic for non-expert in the field. Please explicit why this is relevant to mention (I guess this mean that chemical impact on plant is assumed to be minimal?).
- Figure 2: make sure that structure and size of this figure is optimize for correct reading.
- Line 196-97: "the aim is to train a ML model to capture vegetation impact inferred from multispectral satellite image and explain it…" – as shown in figure 2, the ML algorithm is only used to look at potential parameters explaining pattern of minV and mint, but the detection of the impact (derivation of minV and minT) is not based on ML method.
- Lines 223-224: "we test here…" – consider moving this sentence after the first sentence of the paragraph, as it would clarify the first sentence.
- Equation 1: definition of CDI(i,j,k) is unclear. The summation sign is apply from t=1 to t, which does not make sense, especially as t is not one of the parameter of the equation. I understand that this is a cumulative indicators that consider the difference in VI before the time

considered, but the equation needs to be reformulated to properly designate the integration over time. As both k and j refer to time somehow, it seems that the summation sign should be applied to both these parameters. This equation needs to be reformulated. The parameter (j,k) are no more used in the graph of figure 3, which would make sense for the year (k) but not for season (j).

- Equation 1 and discussion (line 539, 633-34, 676): more generally, it should be reflected why the CDI index is an absolute difference between the considered vegetation index (EVI) and the value of that index before the eruption, and not a relative difference as is the case for many remote sensing indexes. As observed in the results, the use of the absolute difference leads to the fact that larger difference can only be obtained for areas with initially high VI values, leading EVI to be the main controlling factors of the CDI variation.
- Lines 244: 'where CDI(I,j,k) remains negative' – I think you here refer to the trends/slope in CDI and not its absolute value (in scenario 2 and 3, CDI also remains negative on figure 3).
- Line 253-54: "the shape of the CDI curve after reaching minV is not considered here" – this is really a pity as this potentially provide a lot of information on the recovery process. Although I understand that this requires to define additional characteristic index for the CDI curve, authors might reflect on the potential of characterizing recovery in their discussion.
- Figure 2 and following text and figures: the term 'Fall3d' and 'Fall3D" are used throughout the manuscript; please homogenize.
- Lines 285: why do you use Fall3D to model tephra dispersal and not other commonly used model (Tephra2)? Please justify.
- Line 301-306: the definition of climatological variables and the equation used for SRI is cryptic. The assumptions/hypotheses behind these definition should be made explicit. It is unclear why only values prior to the eruption are used, but still the temporal aspect is considered. What is the idea behind considering different time periods of 1, 2, 3, 6 and 12 months? What is the SRI trying to assess: is it related to the interannual variability or the seasonal variability within a year? Again, what would be the relevance of considering inter-annual variability of rainfall before the eruption (do you assume that inter-annual variability measured over 5 years is characteristics for the area, and can be expected to also apply during after the eruption)? This issue is mentioned in lines 645-46, but without clear justification.
- Section 3.2.4 – land cover: you mention that land cover data is available yearly from 2015-19 but do not mention which year is used as reference in your analysis. In addition, in the limitation, you do not mention that using post-eruption land cover data might be a cause of bias, as in extreme case one could expect land cover to have been modified on the long term due to the eruption (e.g. abandoned agricultural land, non-recovered forest, …)
- Line 333 (and other): 'one hot encoding' – I personally do not know what this means. At least add a reference for non expert to understand what this is.
- Line 345: cut that sentence in two sentences.
- Lines 381: 'as generalization is not the main purpose of this study, SHAP values are computed on the full dataset…" – this sentence is unclear and require further explanation if important.
- Figure 6 is a great figure and helps to understand the CDI concept.
- Section 4.2: consider organizing such long section (and similar ones) into logical paragraphs.
- Line 452: 'local CDI minimum about a year after the eruption" – as pointed out by the authors themselves later on, this is only true for a smoothing of 1 month, not for a smoothing of 3 months, in which case the trends continue downward. As authors finally decide to select the 3 month smoothing, this statement should be revised.

- Line 497: "no clear relationships appears between minT and either minV or tephra load" – I do not agree with this statement. At least the very high minT (>1500 days) value are clearly related to larger minV and higher ash load.
- Lines 510-511: the alpha, lambda and min child weight parameters have not been introduced when presenting the ML model. This is not understandable.
- Lines 550-51: 'this suggests that forest are potentially more resilient…" – although what you suggest here seems rational, I do not see how this is being derived from the results of the model. Please explain.
- Abiotic factors: authors group the climatic factors into the terminology 'abiotic', although this includes temperature and precipitation integrated over different time periods; little explanation is provided for why specific variable are relevant in one case and not in another. Interpretation of the role of temperature would be probably quite different than that of precipitation.
- Line 578 and other similar occurrences: authors refer several times to 'stepwise' variation in the Shap graphs. Although some of these 'tresholds' are clear, many of them are not so clear, and their identification should be subject to caution. In Figure 8a, only the step >550 kg/m² is very clear. Others steps should be evidenced by trend lines or other visual representations if they can be statistically supported.
- Lines 615 : rephrase this sentence; reference to 'most negative' and 'both positive and negative behaviour' in the same sentence is unclear.
- Lines 636-38: no explanation is given for the strong role of 'elevation' in many of the observed trends. More generally, no attention is pointed out to the potential strong correlation between elevation and many of the other controlling variables (temperature, precipitation, land cover, …). How would these correlations influence the outcome of the model? Is there not a need to check for cross-correlations and avoid some variables from the model to help interpretation?
- Lines 660-65: the concept of precipitation anomaly is insufficient clear to understand what these observations mean. How could precipitation variability before eruption be of any relevance to explain vegetation recovery after the eruption? Why would this be specifically relevant for crop areas? Explain.
- Line 703 and lines 720-21: 'focus on impact mechanism' – as pointed correctly by authors further in the discussion (lines 776-780), the ML results highlight statistical links but the causal relationships (and therefore mechanism) cannot directly be extracted from the results, and would need to be complemented with other sets of evidence. Especially as several of the controlling variables are strongly correlated with each other. Caution is therefore needed in this claim.
- Line 712-15: check sentence structure, it misses a verb.
- Lines 781-84: as mentioned in previous comments, the use of pre-eruptive co-variates and their definition (especially precipitation and temperature) is not clear throughout the manuscript. The reason for using these pre-eruptive data and for considering them as relevant should be better explained. How post-eruptive dataset could be integrated could also be shortly mentioned.
- Conclusions: the innovative character of the research could be made more explicit by referring to relevant research considering other processes impacting on vegetation (fires, drought, floods…) and whether the proposed method would be applicable (or has already been applied) to tackle thee processes. The practical implication of the findings (and further application of the proposed approach) for volcanic risk assessment and management could also be shortly mentioned.

---

## Author Response (AR1)

**Reviewer 1**

We are grateful to Reviewer 1 for his constructive and detailed reviews, his enthusiasm for the method and for his perspicacity in picking some important flaws of our manuscript (e.g., the formalization of Equation 1). Critical comments included:

1. A better definition of climatic pre-eruption variables in the model;
2. A clarification of multicollinearity between features;
3. Limitations of model inference.

We have answered below each comment in a detailed way, which we believe addresses all of Reviewer 1's comments.

**Minor comments:**

- **Line 71**: Removed « five ».
- **Figure 1a**: We added labels where possible. However, labeling the innermost isopach is obscuring key features of the proximal deposit.
- **Line 172-174**: The sentence was rephrased.
- **Figure 2**: The figure was reworked.
- **Line 196-197:** Yes, we modified the text following this suggestion.
- **Lines 223-224**: We modified the text following this suggestion.
- **Equation 1**: Thank you for pointing that. In fact, expressing the CDI in a formal mathematical way is not trivial. We propose a new version of Eq. 1 and proper referenced its indices throughout the text.
- **Equation 1** and discussion: We have made this point clearer when introducing the CDI (Section 3.1.2).
- **Line 244**: This was corrected.
- **Line 253-54**: This is an excellent point – and not acknowledging this limitation is clearly an omission from our part. This is indeed a critical aspect that will be investigated in future iterations of the model once a satisfactory modeling method has been identified to account for multi-target predictions. We added clarifications just before and within the *Caveats and future research* section.
- **Line 285**: The use of steady-state analytical models such as Tephra2 is unsuitable for modeling a month-long eruption. We do not feel a justification is required.
- **Line 301-306** : We added clarifications in Section 3.2.2.
- **Section 3.2.4** – land cover: The year is indeed specified (i.e., 2015). We have added a comment addressing the limitation raised by the reviewer.
- **Line 333**: We removed any reference to *One-Hot Encoding.*
- **Line 381**: This is indeed very important and has been rephrased.
- **Line 452**: Done
- **Line 497**: The statement has been modified.
- **Lines 510-511**: It is difficult to explain the fundamentals of gradient-boosted trees in the manuscript. Therefore, we have removed this sentence from the text, but we added a short description of each parameter in the caption of Table 4.
- **Line 550-551**: This was removed as it is better explained further in the text.
- We source the use of *abiotic* vs *biotic* factors from existing literature (e.g., Arnalds, 2013). Note that although *abiotic* factors are commonly restricted to environmental parameters, we choose to keep *abiotic* factor to also englobe socio-economic components of agriculture (and especially crops) vulnerability. In addition, as described

in the discussion, we restrict here any causal inference "*to effects that rely on phenomena that have been either witnessed in the field or experiments*". In this sense, we agree that temperature and precipitation most likely have different roles in explaining the impact, but we are currently reluctant to *overinterpret* the results of our model.

- **L578**: About 2'500 different combinations of SHAP dependence plots can be produced. Although we focus here on 19 of them, the results highlighted in the text are based on an exhaustive review of a majority of them. It is however impractical to include and describe all of them in the text. Although we agree that their interpretation requires caution, we have adopted the most conservative interpretation when suggesting noticeable patterns in our data and only highlight them when they agree with other sources of information (e.g., post-EIA). In this sense, we have modified the text to only use conservative phrasing (e.g., "suggest" rather than "show") to stress the care required in inferring an unrealistic degree of causation.
- **Line 615**: Rephrased.
- **Line 636-638**: Thank you for pointing the role of elevation in our model. Firstly we have added more details in the discussion section that addressed this spatial dependency (Caveats and future research section). Secondly, we are facing here a problem of multicollinearity rather than simple correlations. As explained in Section 3.2, we have reduced an initial dataset of ~300 variables to about ~40 based on standard exploratory data procedures aiming at i) reducing the amount of collinearity between features and ii) improving the model's prediction by identifying and removing uninformative variables. This procedure is highly iterative and somehow standard, which is why we have decided not to include it into details in the manuscript. We would however like to point that elevation and landcover were not overly correlated. We have added clarifications in Section 3.2. Thirdly, one benefit of the *XGBoost* library and gradient-boosted trees compared to other decision trees is their ability to handle multicollinearity (we have added clarifications and associated references in Section 3.4.1). Finally – and linked to the previous point and multiple previous answer, variables in Earth and environmental sciences are rarely purely orthogonal. Again, this is a problem for model inference, which we keep to a minimum. Conversely, elevation proves to be an important variable even when the model is trained on individual landcover classes. This outlines how i) multicollinearity occurs over a variety of variables, which makes the choice of removing specific variables more difficult and ii) despite this, the model remains informative.
- **Line 660-665**: We added clarifications in Section 3.2.2.
- **Line 703/720-721**: We feel this goes back to concerns about multicollinearity and model inference, which we have already address. Please refer to comments for Line 578 and 636-638.
- **Line 712-715**: Rephrased
- **Line 781-784**: The paragraph was remodeled.
- **Conclusions:** A lot could indeed be added in the discussion and the conclusion. However, since the paper is already long and mostly represents an introduction to the method, we decided to focus on pragmatic conclusions.

**Reviewer 2**

**Point 1**

Earth Observation data provides a global observation network in space and time, which can be used as an indirect proxy to infer surface processes. EO sensors always balance three types of resolutions (i.e., spatial, temporal and spectral) to achieve specific purposes. For instance, Landsat provides 30 m pixels at a poor temporal resolution whereas MODIS provides 250 m pixels at a higher frequency. With this in mind, the EO community is entirely aware that satellite imagery is not able to resolve the complexity suggested by Reviewer 2, especially when, in our case, the methodology considers large areas and require dense time series. However, EO data enables the monitoring of vegetation over widespread areas, for which MODIS is often the preferred sensor. Therefore, we feel that i) the manuscript already extensively covers literature focusing on vegetation monitoring from EO data and ii) expressing and detailing this uncertainty is beyond the scope of the paper. Please note that all points stated in our discussion and future objectives target this exact purpose through validations and comparison with field mapping.

**Point 2**

We partly agree with this statement. We agree that impact mechanisms have been discussed for a long time in the literature, both from an ecological perspective (i.e., references proposed by Reviewer 2) and from an "impact" perspective (i.e., the dominance of references proposed in our manuscript), with the conclusion that no consensus is yet possible. One limitation to this is the opportunistic nature of studies in the field, which are too limited (both in number and in spatial coverage) to provide a sufficiently large number of observations required to capture the full variability of the involved processes (e.g., eruption types, climates, crop and vegetation types etc). We would fully agree with Reviewer's 2 comments should our method ambition to replace these field-based studies. However, the motivation for our method is the realization that generalizable models of volcanic impacts – at least for disaster risk reduction perspectives – probably will never be developed using only field-based studies, and we therefore explore here an alternative way to generalize these *in situ* observations rather than replace them. In addition, we fully acknowledge the limitations of our methodology, and limit causal inference to specific case-studies where impact mechanisms suggested by various sources point to supporting our interpretation. We therefore feel that most issues raised by Reviewer 2 in this comment are comprehensively addressed in our manuscript and supported by more recent literature, although using an impact rather than an ecological perspective.

**Point 3**

This statement has been deleted. Please note that i) the timing of the eruption relative to the phenological cycle of the plant is mentioned in the same paragraph, for which we have added relevant references and ii) further investigations of this relationship is identified as the first point for future iterations of the method in the discussion section.

**Point 4**

The method (and the CDI) are indeed based on vegetation indices, which provide a proxy for *biomass production*. Here, we attempt to provide a proxy for *impact*. Following comments of Reviewer 1, the purpose and limitations is now more detailed and discussed in the perspective

of existing techniques. In a nutshell, the CDI is designed to not only capture *negative impacts* but, on the longer term, to also capture the recovery. It was developed with the idea of quantifying impact as a *budget* (i.e., comparing short-term negative losses with potential long-term gains in fertility), and presents many advantages to estimates *rates* of impact and recovery compared to existing anomaly quantification methods. We believe that changes made to address Reviewer 1's comments also address this issue.

**Minor comments**

- Line 32 : Done
- Line 125: The map was reworked according to both reviewer's comments
- Line 175: Done
- Line 190: Done (and good to know!)
- Line 267: Done
- Line 269: Done
- Line 270: This sentence was rephrased following reviewer 1's comments
- Line 310: Done
- Line 800: Done